# Realization of Lieb lattice in covalent-organic frameworks with tunable topology and magnetism

Bin Cui [ID] [1]*, Xingwen Zheng[1], Jianfeng Wang[2], Desheng Liu[1], Shijie Xie[1] & Bing Huang [ID] [2]*

Lieb lattice has been predicted to host various exotic electronic properties due to its unusual Dirac-flat band structure. However, the realization of a Lieb lattice in a real material is still unachievable. Based on tight-binding modeling, we find that the lattice distortion can significantly determine the electronic and topological properties of a Lieb lattice. Importantly, based on first-principles calculations, we predict that the two existing covalent organic frameworks (COFs), i.e., $sp^2$C-COF and $sp^2$N-COF, are actually the first two material realizations of organic-ligand-based Lieb lattice. Interestingly, the $sp^2$C-COF can experience the phase transitions from a paramagnetic state to a ferromagnetic one and then to a Néel antiferromagnetic one, as the carrier doping concentration increases. Our findings not only confirm the first material realization of Lieb lattice in COFs, but also offer a possible way to achieve tunable topology and magnetism in organic lattices.

[1] School of Physics, National Demonstration Center for Experimental Physics Education, Shandong University, Jinan 250100, China. [2] Beijing Computational Science Research Center, Beijing 100193, China. *email: cuibin@sdu.edu.cn; bing.huang@csrc.ac.cn

The electronic properties of a crystal are determined by its crystalline lattice symmetry. Lieb lattice, a two-dimensional (2D) edge-depleted square lattice (Fig. 1a), can be regarded as the reduced 3D perovskite lattice[1]. As one of the most important frustrated lattices, the ideal Lieb lattice can host exotic electronic structures, which is featured by the Dirac cone intersected by a flat band (Dirac-flat bands)[2], as shown in Fig. 1c. Interestingly, various physical phenomena, e.g., topological Mott insulator[3], superconductivity[4], ferromagnetism[5,6], and fractional quantum Hall (FQH)[7–10] effects, have been predicted to exist in the ideal Lieb lattice systems. However, until now only a few Lieb lattices have been realized in artificial lattice systems, e.g., molecular patterning lattices on metal substrates[11,12], photonic and cold-atom lattices[13–16], rather than a real material system, which significantly prevent the realization of these unusual physical properties of Lieb lattice for practice applications.

Although it is found that the Lieb lattice is extremely difficult to be realized in an inorganic material, it is still possible to realize such a lattice in an organic one, as the structural flexibility allows an organic system exhibiting diverse crystalline symmetries[17–25]. Especially, it is noticed that the covalent-organic frameworks (COFs) can host various 2D lattices[18,21,24,26], including square lattice[21,26], which makes the discovery of a Lieb lattice in a COF possible. Usually, the molecular orbitals (MOs) play an essential role in determining the electronic properties of a ligand-based organic semiconductor, e.g., a COF. Since the MOs in an organic system can be more easily modulated than the atomic orbitals (AOs) in an inorganic system by the lattice distortions, it is expected that the tunable electronic and magnetic states could be achieved in an organic Lieb lattice.

In this article, using tight-binding (TB) modeling and first-principles density-functional theory (DFT) calculations, we find that the electronic and topological properties of a Lieb lattice can be effectively modulated by its lattice distortions. Interestingly, we discover that $sp^2$C-COF and $sp^2$N-COF, which are synthesized in recent experiments[27], are the first two material realizations of organic-ligand-based Lieb lattices. The electronic structures of these two COFs around the band edges can be well characterized by the MOs-based Dirac-flat-band models. Furthermore, we find that the lattice distortions can also dramatically affect the bandwidth of the Dirac-flat bands, which in turn determines its electronic instability against spontaneous spin-polarization during carrier doping. Remarkably, it is found that both ferromagnetic (FM) and Néel antiferromagnetic (AFM) phases can be realized in these distorted Lieb lattices under certain hole doping concentrations ($n_h$). Our discovery not only confirms the first material realization of Lieb lattice in COFs, but also suggests an interesting routine to realize tunable topological and magnetic states in $d$-($f$-) orbital-free organic lattices.

## Results

**Three-band Lieb lattice models.** We begin with the three-band tight-binding Hamiltonian of Lieb lattice [without spin-orbit coupling (SOC) effects], including one corner and two edge sites

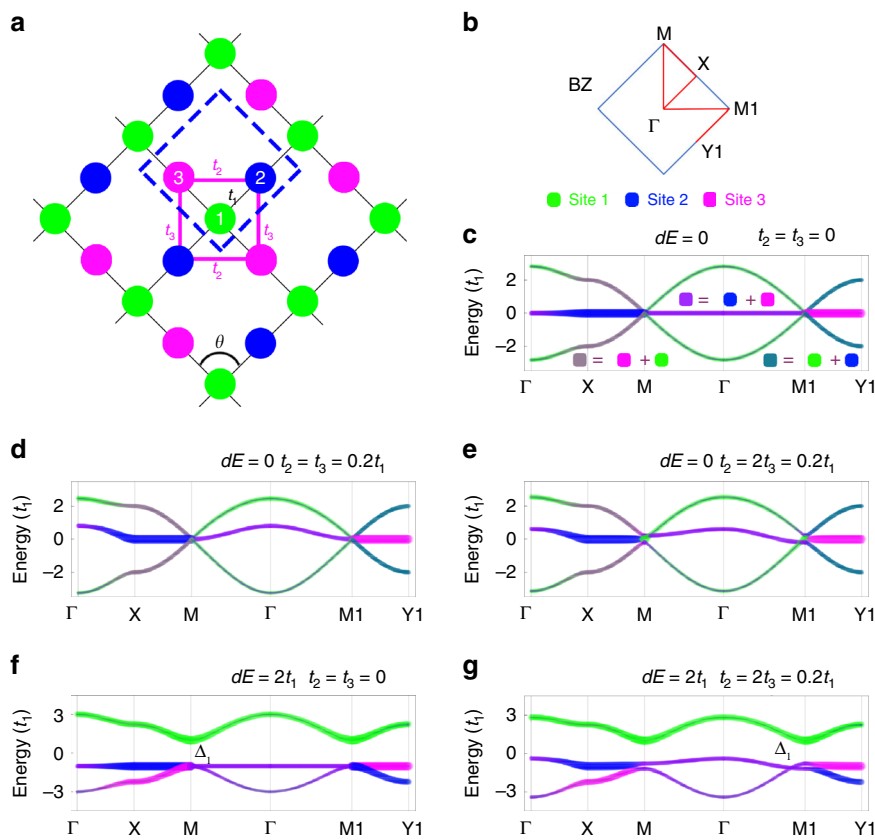

**Fig. 1 Three-band Lieb lattice models. a** Geometrical structure of a Lieb lattice, which contains one corner site 1 and two edge sites 2 and 3 in the unitcell (marked by the blue dashed-lines). Strength of lattice (geometrical) distortion is indicated by the change of structural angle $\theta$ away from 90°. See text for the definitions of other parameters. **b** Brillouin zone of a Lieb lattice. It notes that M and M1, equal in an ideal Lieb lattice, are inequivalent in a distorted Lieb lattice ($\theta \neq 90°$ or $t_2 \neq t_3$). Projected band structures of an ideal Lieb lattice **c** without and **d** with NNN hopping. **e–g** Projected band structures of distorted Lieb lattices under three representative situations. In (**c–g**), the colors of the energy bands represent the wavefunction components from different lattice sites shown in (**a**). Curve width indicates component weight.

in a 2D line-centered square lattice unitcell:

$$H_0 = \sum_i E_i c_i^\dagger c_i + \sum_{\langle i,j \rangle} \left( t_1 c_i^\dagger c_j + h.c. \right) \\ + \sum_{\langle\langle i,j \rangle\rangle} \left( t_{2,3} c_i^\dagger c_j + h.c. \right)$$

(1)

where $c_i$ ($c_i^\dagger$) annihilates (creates) an electron with the energy of $E_i$ on the $i$th site ($i = 1, 2,$ or $3$) in the unitcell, and $t_1$ and $t_{2,3}$ are nearest-neighbor (NN, $\langle i,j \rangle$) and next nearest-neighbor (NNN, $\langle\langle i,j \rangle\rangle$) hopping integrals, respectively, as shown in Fig. 1a. The details of TB modeling can be found in the Supplementary Notes 1–3.

For an ideal Lieb lattice, the on-site energy of the three sites are identical, i.e., $E_1 = dE/2 = 0$ and $E_2 = E_3 = -dE/2 = 0$. Then, we can calculate the (three-band) electronic structure of an ideal Lieb lattice. When the NNN hopping is not included, as shown in Fig. 1c, the calculated band structure is featured by the Dirac cone intersected by a flat band. The wavefunction of flat band is mostly localized at the edge sites, while the wavefunctions of Dirac bands are equally contributed by both corner and edge sites. When NNN hopping is included, e.g., $t_2 = t_3 = 0.2t_1$, the features of the Dirac-flat bands still maintain besides that the flat band becomes more dispersive around the $\Gamma$ point in the Brillouin zone (BZ), as shown in Fig. 1d.

Since the ideal Lieb lattice is very rare in solids[3–10], it is curious for us to further understand the electronic structures of distorted Lieb lattices. Firstly, we consider the situation of $\theta \neq 90°$, e.g., $\theta < 90°$, corresponding to the geometrical lattice distortions. In this case, the fourfold rotation symmetry of the system is no longer valid, then $t_2 \neq t_3$. As shown in Fig. 1e, the Dirac-flat bands decompose into two sets of Dirac cones that deviate away from M (M1) to $\Gamma$ point along the $\Gamma$-M ($\Gamma$-M1) in the BZ. Secondly, we consider the situation of $dE \neq 0$, which corresponding to different atom (ligand) occupations at corner and edge sites in an inorganic (organic) lattice. In this case, the band degeneracy of Dirac-flat bands is broken and a bandgap $\Delta_1$ (hereafter, $\Delta_1$ is defined as the bandgap between the top Dirac and middle flat bands) is induced, as shown in Fig. 1f. Meanwhile, the flat band and the bottom Dirac band still keep in touch with each other. Interestingly, it is found the wavefunction components of these Dirac-flat bands are dramatically different from the ideal Lieb lattice, i.e., the top Dirac band is contributed by the corner sites while the flat and bottom Dirac bands are contributed by the edge sites. Finally, we consider the distortions with both $\theta \neq 90°$ and $dE \neq 0$, as shown in Fig. 1g, which correspond to the realistic situations in many materials, including the COFs focused in our study. In this case, the changes of Dirac-flat bands could be considered as a combined effect of $\theta \neq 90°$ (Fig. 1e) and $dE \neq 0$ (Fig. 1f). Interestingly, only one Dirac cone can survive around M1 point along the $\Gamma - M1$ line.

**Distorted Lieb lattice model with SOC effects.** We further consider the SOC effects on the distorted Lieb lattice model, with the total Hamiltonian $H = H_0 + H_{SO}$. $H_{SO}$ is considered as an imaginary hopping between the NNN sites, similar to Kane and Mele's SOC term[9,28]:

$$H_{SO} = -i\lambda \sum_{\langle\langle i,j \rangle\rangle} \left( \mathbf{d}_{ik} \times \mathbf{d}_{kj} \right) \cdot \mathbf{s}_{\alpha\beta}^z c_{i\alpha}^\dagger c_{j\beta}$$

(2)

where $\lambda$ is the SOC constant and $\mathbf{d}_{ik}$ ($\mathbf{d}_{kj}$) denotes the unit vector from edge (corner) site $i$ ($k$) to corner (edge) site $k$ ($j$) (see Supplementary Fig. 1). $\mathbf{s}$ is the Pauli matrix representing the electron spin. Generally, we find that SOC effects can lift the degeneracy of the Dirac cone, and the topological properties of these Dirac-flat bands strongly depend on the value of $dE$ (the values of other parameters are the same as those in Fig. 1g).

For $dE = 0$ (Fig. 2a), the bandgaps of $\Delta_1$ and $\Delta_2$ (hereafter, $\Delta_2$ is defined as bandgap between middle flat and bottom Dirac bands) can be induced around M and M1 points. It is noted that $\Delta_2$ can only be induced by SOC effects. For the case of $dE \neq 0$, we find that there is a competing mechanism between $dE$ and SOC for the changes of $\Delta_1$. When $0 < dE < dE_c$ ($dE_c = 2\sqrt{(t_2 - t_3)^2 + 4\lambda^2}$, see Supplementary Notes 3–5), $\Delta_1$ decreases as $dE$ increases. When $dE = dE_c$, $\Delta_1 = 0$, as shown in Fig. 2b, two Dirac cones can form around M and M1 points. When $dE > dE_c$ (Fig. 2c), $\Delta_1$ opens again and increases as $dE$ increases, which is opposite to the case of $dE < dE_c$. Meanwhile, when $dE$ changes through $dE_c$, the wavefunction compositions of top Dirac band and middle flat band are switched, as shown in Fig. 2a, c.

The topological properties of each band can be characterized by its spin Chern number, due to the spin degeneracy and conservation of $S_z$. The spin Chern number for the $n$th band is defined as $C_n^s = \left( C_{n\uparrow} - C_{n\downarrow} \right)/2$, and $C_{n\sigma}$ is Chern number for the spin-$\sigma$ ($\sigma = \uparrow, \downarrow$) of the $n$th band, which can be calculated from the integral of the Berry curvature over the whole BZ[29,30] (see "Method" section and supplementary Fig. 2). As shown in Fig. 2, when $dE = 0$, the calculated $C_1^s = -1$ and $C_3^s = +1$ for the top and bottom Dirac bands, respectively, and the calculated $C_2^s = 0$ for middle flat band. Interestingly, when $dE > dE_c$, the top Dirac and middle flat bands can switch their topologies, as shown in Fig. 2c, i.e., $C_1^s = 0$ ($C_2^s = -1$) for the top Dirac (middle flat) band.

The nontrivial topology of these Dirac-flat bands in Fig. 2a–c can be reflected by edge states calculations, as shown in the according Fig. 2d–f. It can be seen that the helical edge states with opposite spin channels connect the two bands with nonzero $C_n^s$, indicating the quantum spin Hall effect.

It is emphasized that in a realistic material, only the total spin Chern number $C_s$, the sum of $C_n^s$ for all the occupied bands, is associated with the observable quantum conductance of the system. Therefore, the topological properties of a realistic Lieb lattice material sensitively depend on the position of Fermi level, and the charge doping may be needed to achieve a nontrivial state.

**Realization of distorted Lieb lattices in COFs.** Taking advantage of the structural flexibility of organic materials, numerous COFs with various crystalline symmetries have been synthesized in recent years. Remarkably, after extensive structural analysis, we notice that two COFs with $sp^2$ hybridized backbones fabricated in the very recent experiments, named as $sp^2$C-COF and $sp^2$N-COF, can realize slightly distorted square Lieb lattices[27] ($\theta < 90°$), as shown in Fig. 3a, b, respectively. Meanwhile, the ligands at the corner and edge sites are different in monolayer $sp^2$C-COF and $sp^2$N-COF, corresponding to the case of $dE \neq 0$. In both COFs, a pyrene surrounded by four phenyl rings is located at the corner site, named as 1,3,6,8-tetrakis(2-phenyl)pyrene (TPPy), while the ligands at the edge sites in these two COFs are slightly different, i.e., 1,4-bis(2-crylonitrile)benzene (BCNB) in $sp^2$C-COF and 1,4-bis(2-aldimine)benzene (BAIB) in $sp^2$N-COF. Meanwhile, crylonitriles (CN) and aldimine (AI) bond to the $bis$- positions on the phenyl ring of edge sites in $sp^2$C-COF and $sp^2$N-COF, respectively. The enlarged view of connection parts between corner and edge sites are shown in Supplementary Fig. 3.

We have calculated the electronic properties of monolayer $sp^2$C-COF and $sp^2$N-COF using DFT-PBE calculations, as shown in Fig. 3c, d, while the results for bulk $sp^2$C-COF and $sp^2$N-COF can be found in Supplementary Fig. 4. Generally, it is found that monolayer and its corresponding bulk COFs have similar electronic properties due to the relatively weak interlayer van der Waals (vdW) interactions. The bandgaps of monolayer $sp^2$C-COF

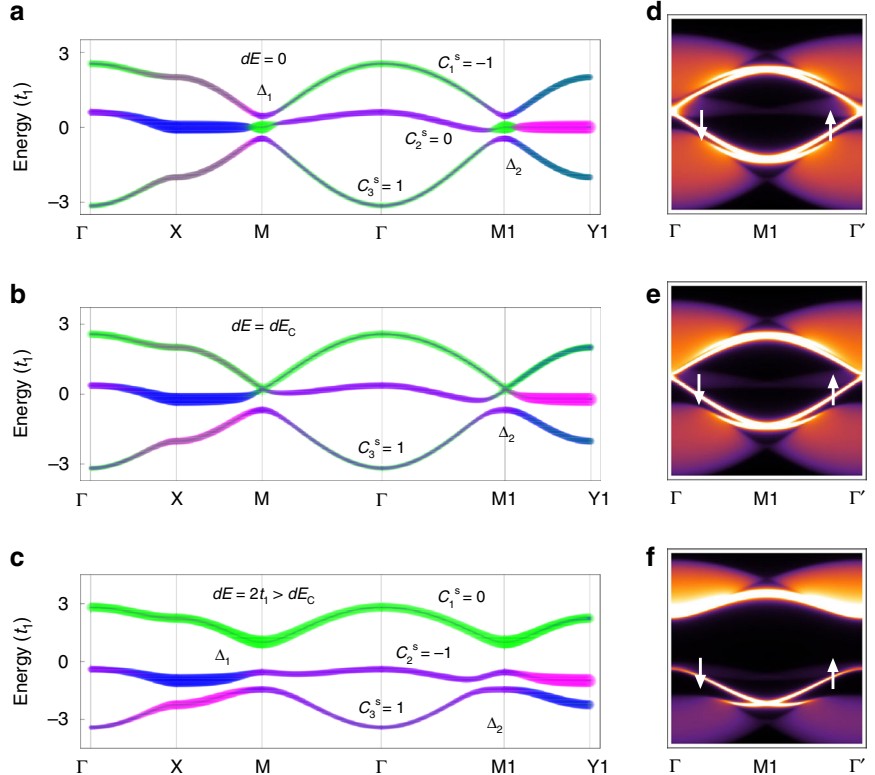

**Fig. 2 Distorted Lieb lattice models with SOC effects. a–c** Calculated band structures of distorted Lieb lattices with SOC effects ($\lambda = 0.1 t_1$). TB parameters of distorted Lieb lattice are adopted from Fig. 1g but with variable $dE$: (**a**) $dE = 0 < dE_c$, (**b**) $dE = dE_c$, and (**c**) $dE = 2t_1 > dE_c$. $dE_c = 2\sqrt{(t_2 - t_3)^2 + 4\lambda^2}$. Calculated spin Chern number for each band ($C_n^s$) is also labeled. **d–f** Spin-resolved edge states for each band structure in (**a–c**), respectively.

and $sp^2$N-COF are 1.45 and 1.30 eV, respectively, which is smaller than the experimental values (e.g., 1.9 eV for $sp^2$C-COF)[27] due to the underestimation of bandgaps in DFT calculations.

Here, we mainly pay our attention to the band edges, especially for the top three valence bands, i.e., VB1–VB3 marked in Fig. 3c, d, in both COFs. In the band structure of $sp^2$C-COF (Fig. 3c), VB1 separates from VB2 and VB3 in energy, opening an energy gap of $\Delta_1 \sim 0.4$ eV. Notably, VB2 and VB3 touch with each other forming a single Dirac cone near the M1 point on the $\Gamma - M1$ line. The wavefunction distributions of energy bands can be better understood by the ligand-based MOs than AOs in a COF. Generally, the band edges in a ligand lattice are formed by the HOMOs and LUMOs of the ligands, respectively[31], and the interaction between the HOMOs and LUMOs are negligible. Therefore, the VB1–VB3 can be considered separately from the CB1–CB3 in a TB modeling based on the Lowdin perturbation theory. As shown in Fig. 3c, the charge densities of VB1–VB3 are projected into the ligands of $sp^2$C-COF. Interestingly, it is found that the top VB1 is solely contributed by the HOMO of TPPy (corner ligands), while the less dispersive VB2 and the bottom VB3 are contributed by the HOMOs of both BCNBs (edge ligands), respectively. Interestingly, it is found that the band dispersions and wavefunction distributions of VB1–VB3 (Fig. 3c) indeed agree well with the three-band distorted Lieb lattice model under the situation of $dE \neq 0$ and $\theta \neq 90°$ (Fig. 1g). The band structure of $sp^2$N-COF (Fig. 3d) is similar to that of $sp^2$C-COF. The $\Delta_1$ in $sp^2$N-COF is ~0.2 eV, which is smaller than that in $sp^2$C-COF. Meanwhile, it is also noted that VB1 in $sp^2$C-COF is much less dispersive than that in $sp^2$N-COF. The calculated SOC effects ($\Delta_2 \sim 0.03$ meV) are negligible in these two COFs, indicating that it could be challenging to observe the nontrivial topological properties (Fig. 2) in these two COFs.

Since VB1–VB3 in these two COFs are similar to that of three-band distorted Lieb lattice model, we can further understand the band dispersions of VB1–VB3 using TB band fitting of DFT calculations. Overall, we find that the differences of VB1–VB3 band dispersions in these two COFs are mainly determined by the difference of $dE$ between the HOMOs of corner and edge sites in these two systems. It is found that $dE$ determines the size of $\Delta_1$, i.e., the larger the $dE$, the larger the $\Delta_1$ will be (without SOC effects). Meanwhile, $\Delta_1$ can in turn determine the band dispersion of VB1, i.e., the larger the $\Delta_1$, the flatter the VB1 will be. In addition, the interfacial torsion angle between the corner and edge ligands in the $sp^2$C-COF is larger than that in the $sp^2$N-COF (see Supplementary Fig. 3b, e). The larger torsion angle can reduce the conjugation in a larger degree in the COF and lead to a smaller $t_1$. Therefore, the larger $dE$ and torsion angles in $sp^2$C-COF than in $sp^2$N-COF can give rise to a flatter VB1 in $sp^2$C-COF, agreeing well with the DFT results. On the other hand, our understandings suggest a possible way to design flat bands in organic lattices, as the flat bands could have great importance for realizing many novel physical phenomena.

Similarly, the bottom three conduction bands (CB1–CB3), formed by the LUMOs of the edge and corner ligands, can also be understood by the MOs-based Lieb lattice model. The TB band fitting of CB1–CB3 within Lieb lattice model is shown in Supplementary Fig. 5. Due to the small $dE$ between the LUMOs of corner and edge ligands, the CB1–CB3 formed Dirac-flat bands (Fig. 3c, d) are close to the ideal ones (Fig. 1c).

**Ferromagnetism in $sp^2$C-COF and $sp^2$N-COF.** In the experiments, the FM phase, with an easy axis perpendicular to the COF plane, i.e., along the $z$ direction in Fig. 3a, b, is found in $sp^2$C-

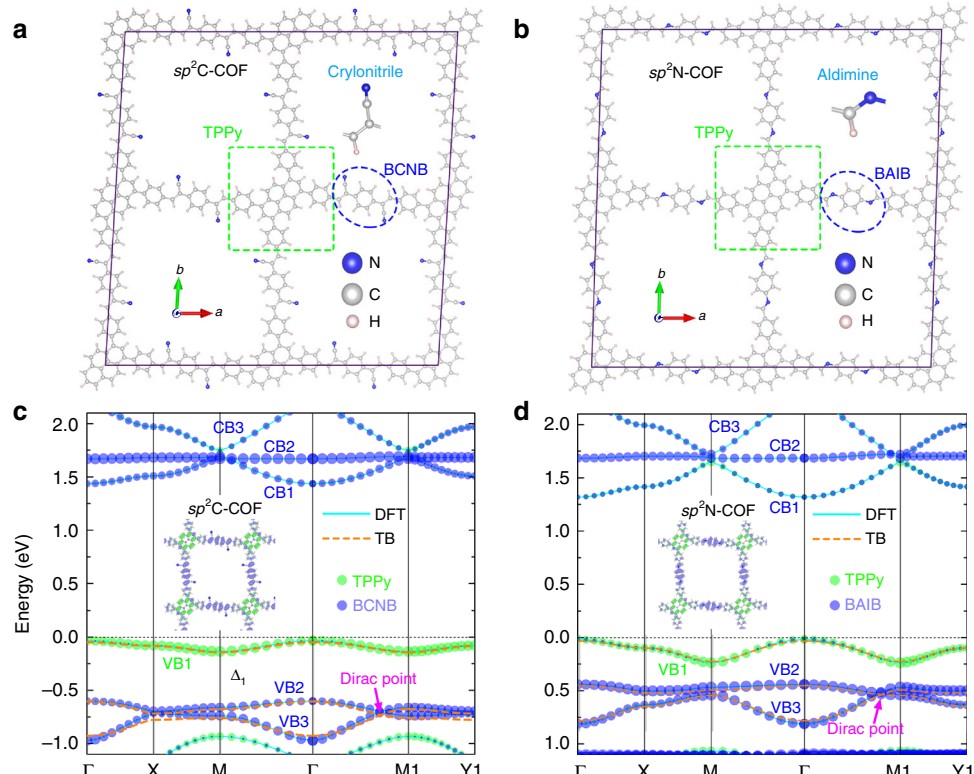

**Fig. 3 Realization of distorted Lieb lattices in COFs.** Top views of (**a**) $sp^2$C-COF and (**b**) $sp^2$N-COF in a 2 × 2 supercell. The corner (TPPy) and edge (BCNB or BAIB) ligands are marked by the green-dashed squares and blue-dashed ellipses, respectively. Insets: Enlarged structures of end groups on the edge sites, i.e., crylonitrile for BCNB and aldimine for BAIB. **c** and **d** Projected band structures of monolayer $sp^2$C-COF and $sp^2$N-COF, respectively, where the green and blue circles (the sizes represent the weight of MOs) indicate that the states are contributed by corner (TPPy) and edge (BCNB or BAIB) sites, respectively. Insets: Partial charge distributions of top three valence bands. Orange dashed-lines in (**c**) and (**d**): TB fittings of DFT band structures based on distorted Lieb lattice model for these top three valence bands. In (**c**) $sp^2$C-COF: $t_1 = 0.1$ eV, $t_2 = 0.4t_1$, $t_3 = 0.45t_2$, $dE = 5.9t_1$; in (**d**) sp2C-COF: $t_1 = 0.12$ eV, $t_2 = 0.2t_1$, $t_3 = 0.45t_2$, $dE = 2.5t_1$.

COF after iodine oxidization at a low temperature (8.1 K)[27]. This is unusual as the $sp^2$C-COF not containing any transition-metal atoms can evoke FM ordering, which inspires us to further understand the magnetic properties of these COFs after iodine doping. Our extensive test calculations indicate that the most important role of iodine doping is to introduce extra holes in $sp^2$C-COF and $sp^2$N-COF. The $n_h$ depends on the iodine concentration, but it is insensitive to the adsorption sites of iodine atoms. Besides of the role of hole doping, ionized iodine dopants can introduce some localized defect levels around $E_F$. In the following, we will discuss the mechanism of $n_h$-dependent magnetic phases in the main text and leave the results of iodine doping in Supplementary Fig. 6. According to the Mermin–Wagner theorem[32], for a 2D system there is no realistic long-range order at finite temperature. Hence our discussion should be constrained on finite-size 2D lattices.

Figure 4 shows the band structures of COFs under a typical doping concentration of $n_h = 0.5$ holes per unitcell (u.c.). The VB1 in both $sp^2$C-COF and $sp^2$N-COF can become partially occupied. According to the Stoner model[33], the FM of iterative carriers can arise from partially occupied (except for the case of half-filling[34]) band with a sufficient narrow bandwidth, i.e., the smaller the bandwidth is, the larger the effective electronic interactions will be. Since VB1 in $sp^2$C-COF has a rather narrower bandwidth than that in $sp^2$N-COF, the Stoner criterion for realizing a FM ordering can be more easily satisfied in $sp^2$C-COF than in $sp^2$N-COF. It is found that the spontaneous spin polarization exists in both COFs under $n_h = 0.5$ holes per u.c., and the COFs becomes metallic. The calculated exchange energy

splitting of VB1 in $sp^2$C-COF ($sp^2$N-COF) is about 85 (40) meV. The spin density distributions of these two systems are mainly localized around the corner sites (see Supplementary Fig. 7). For the $sp^2$C-COF, the magnetic moment is 0.5 $\mu_B$ per u.c. at $n_h = 0.5$ holes per u.c., i.e., 1.0 $\mu_B$ per hole. The FM state is energetically more favorable than the AFM one by $\Delta E = E_{FM} - E_{AFM} = -5.0$ meV [or than the nonmagnetic (NM) one by $-6.9$ meV] (see Supplementary Table 1). For $sp^2$N-COF, the magnetic moment is ~0.378 $\mu_B$ per unitcell and the FM ordering is slightly more stable than the AFM one ($\Delta E = -1.5$ meV). Therefore, the stronger FM stability in $sp^2$C-COF is induced by its narrower VB1 bandwidth, which originates from the larger $dE$ and torsion angles in $sp^2$C-COF than in $sp^2$N-COF.

Although the magnetism originates from the many-body effects, which are usually underestimated by the mean-field approximation (MFA) within the DFT methods, the spin-polarized ground states of similar weakly-correlated systems have been correctly evaluated[35–41] and confirmed in the experiments[42,43]. The onsite electron-electron (e–e) interactions ($U$) in the $sp^2$C-COF and $sp^2$N-COF ($sp^2$-hybridized C and N systems) are very weak, compared to the traditional magnetic (3d- or 4f-) systems. On the other hand, the widely acceptable remedy within the DFT formalism to include the electron correlations is DFT + $U$ method. We have performed a GGA + $U$ calculation with $U = 1.5$ eV on the 2p orbitals of C and N atoms, and our calculations show that the enhanced $U$ can slightly increase the exchange splitting $E_s$ (see Supplementary Fig. 8) and hence further enhance the stability of FM ground-states. Therefore, the conclusion of FM ordering in hole-doped COFs obtained from DFT calculations can be enhanced under DFT + $U$ calculations.

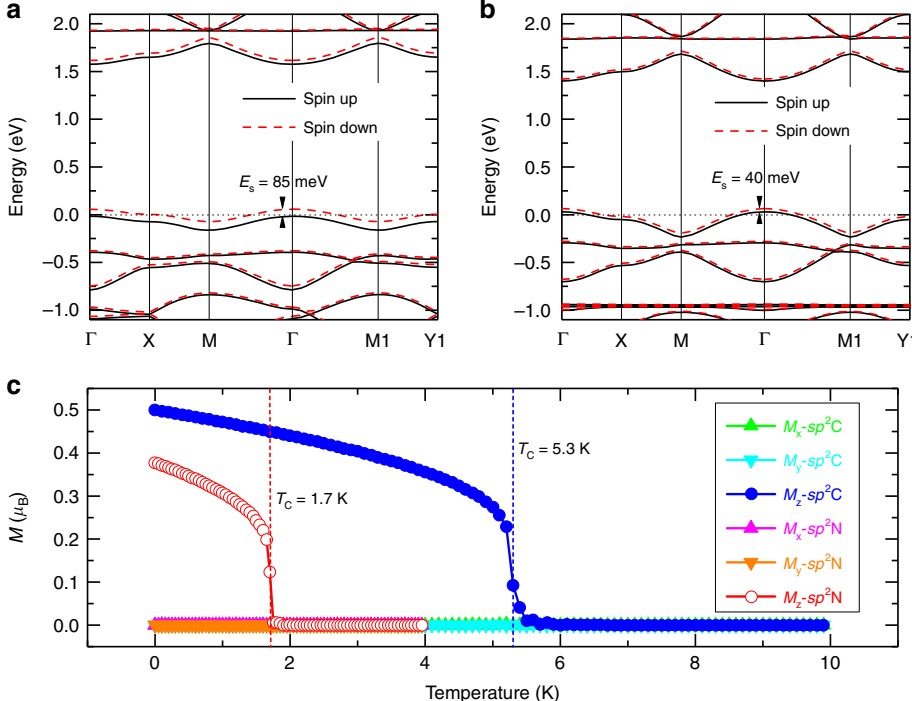

**Fig. 4 Ferromagnetism in $sp^2$C-COF and $sp^2$N-COF.** DFT-calculated spin-polarized band structures of monolayer (**a**) $sp^2$C-COF and (**b**) $sp^2$N-COF with $n_h = 0.5$ holes per u.c. DFT-calculated exchange energy splitting ($E_s$) is 85 meV and 40 meV for $sp^2$C-COF and $sp^2$N-COF, respectively. **c** Calculated average magnetization per site as a function of temperature for monolayer $sp^2$C-COF and $sp^2$N-COF by Monto Carlo simulations. The transition temperatures ($T_c$) are marked with dashed vertical lines.

To estimate the FM transition temperature ($T_c$), we have performed Monte Carlo (MC) simulations within the 2D Heisenberg model with uniaxial anisotropy (See "Methods" section for details), which is successfully applied to simulate the $T_c$ of many 2D systems[44,45]. The reduced exchange integral $J$ and magnetic moment $m$ are 2.5 (1.31) meV·$\mu_B^{-2}$ and 0.5 (0.378) $\mu_B$ for $sp^2$C-COF ($sp^2$N-COF), respectively. The magnetic anisotropic energy (MAE = $E_{\text{out-of-plane}} - E_{\text{in-plane}}$), determined by the energy difference between the out-of-plane and in-plane spin configurations, is ~−0.1 meV for both COFs, which indicates that the magnetic easy axis is perpendicular to the COF planes, agreeing with the experimental observations. Although the value of MAE is small, it is very important for these two COFs to overcome the spin fluctuation and maintain a long-range magnetic ordering at finite temperature. The calculated $T$-dependent magnetic moment (per TPPy) is shown in Fig. 4c. Here, both $sp^2$C-COF and $sp^2$N-COF are under $n_h = 0.5$ holes per u.c. It can be seen that the in-plane components of magnetization are zero at arbitrary temperature for both COFs, but their out-of-plane components are nonzero at low temperature. The calculated $T_c$ is 5.3 K (close to 8.1 K of experiments) and 1.7 K for $sp^2$C-COF and $sp^2$N-COF, respectively.

**Magnetic phase transitions in $sp^2$C-COF.** Besides of $n_h = 0.5$ holes per u.c., we have further explored the possible magnetic phase transitions as a function of $n_h$ at the DFT-level calculations. As shown in Fig. 3, the VB1 of $sp^2$C-COF is contributed by the MO of corner ligands, which has a small effective inter-site (ligand) hopping $t_1 = 0.1$ eV. Meanwhile, the effective onsite (ligand) $U$ is usually in the range of 0.5~1.5 eV for organic systems[46,47], which is significantly larger than $t_1$. The calculated $\Delta E$ as a function of $n_h$ is shown in Fig. 5a. When $n_h < 0.3$ holes per u.c. (yellow region), the magnetic moment on each ligand is too small to evoke a preferred magnetic ordering, giving rise to a paramagnetic (PM) phase. When $0.3 < n_h < 0.75$ holes per u.c., the FM state becomes to be more favorable ($\Delta E < 0$, green region) according to the Stoner's criterion. Interestingly, when $\Delta E$ reaches a maximum value of −5.7 meV at $n_h = 0.55$ hole per u.c., it becomes to be reduced gradually and finally reaches $\Delta E = 0$ at $n_h = 0.75$ holes per u.c. When $0.75 < n_h < 1.0$ holes per u.c., the effect of Fermi surface nesting plays a dominated role in determining its magnetic ground-state, as shown in Supplementary Fig. 9, giving rise to a Néel AFM state (blue region).

Employing the Monte Carlo simulations within the 2D Heisenberg model with the uniaxial anisotropy, we further show $n_h$ dependent magnetic phase diagram of $sp^2$C-COF in Fig. 5b. It can be seen that the FM or AFM phase can survive at finite temperatures under different $n_h$. Interestingly, our calculated magnetic phase diagram of $sp^2$C-COF system agrees well with that by J. E. Hirsch based on a similar model system (at the MFA-level calculations) with $6t_1 < U < 10t_1$[34]. Remarkably, the $T_c$ of FM ordering can reach a maximum value of ~5.7 K for the $sp^2$C-COF at $n_h \sim 0.55$ holes per u.c. Finally, we predict that at an extremely heavy hole doping concentration ($n_h = 3.0$ holes per u.c.), the system can reach a half-metallic Dirac semimetal state, as shown in Supplementary Fig. 10.

## Discussion

t is emphasized that a Heisenberg model is adopted to simulate the $T_c$ of various magnetic phases. Due to finite-size effect, a series of phase transition temperatures can be obtained. Taking such temperature as a proxy to a real $T_c$, we can conclude that the critical temperature of $sp^2$C-COF is higher than that of $sp^2$N-COF. Moreover, the more accurate phase diagram may need more complex methods beyond MFA, e.g., quantum Monte Carlo[48,49] or dynamical mean-field theory[50–52], which is out of the scope of the current study.

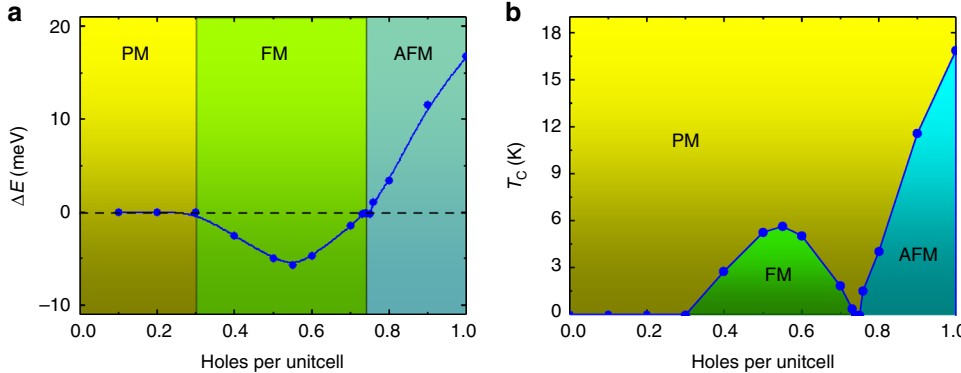

**Fig. 5 Magnetic phase transitions as a function of hole doping concentrations. a** DFT-calculated $\Delta E(\Delta E = E_{FM} - E_{AFM})$ as a function of $n_h$ for monolayer $sp^2$C-COF. **b** $n_h$-dependent magnetic phase diagram for monolayer $sp^2$C-COF, calculated by the MC simulations with 2D Heisenberg models with the uniaxial anisotropy.

In conclusion, based on TB modeling and first-principles calculations, we predict that the $sp^2$C-COF and $sp^2$N-COF, which have been synthesized in the recent experiments, are actually the first two material realizations of organic-ligand-based Lieb lattices. The lattice distortion and inequivalent corner and edge sites make the electronic structures of these two systems deviate from the ideal Dirac-flat bands in a Lieb lattice. Interestingly, these two COF systems can be converted to the FM and AFM states at certain $n_h$ due to the less dispersive top valence bands. Our discovery not only can extend our understanding on the electronic and topological properties of distorted Lieb lattices, but also offers a possible way to realize exotic magnetic states in frustrated organic lattices, which opens the possibility to design the novel organic spintronic devices.

## Methods

**DFT calculations**. Our ab initio calculations for the electronic structures of periodic structures were carried out within the framework of the Perdew-Burke-Ernzerhof generalized gradient approximation (PBE-GGA)[53], as embedded in the Vienna ab initio simulation package code[54,55]. All the calculations were performed with a plane-wave cutoff energy of 450 eV. For the $sp^2$C-COF, we adopted the experimental lattice constants as starting structure to perform geometric optimization. $sp^2$N-COF holds a very similar structure as $sp^2$C-COF based on our calculation. Both COFs were fully relaxed without any constraint until the force on each atom was <0.01 eV Å$^{-1}$. The final lattice constants are $a = b = 24.634$ Å, $c = 3.569$ Å, $\alpha = 79.407°$, $\beta = 100.593°$ and $\gamma = 88.299°$ for the $sp^2$C-COF and $a = b = 24.282$ Å, $c = 3.465$ Å, $\alpha = 78.492°$, $\beta = 101.508°$ and $\gamma = 88.490°$ for $sp^2$N-COF. To eliminate the interlayer interaction, we introduced a vacuum layer of 18 Å thickness for monolayer calculations. The Brillouin zone k-point sampling was set with a $3 \times 3 \times 13$ mesh for bulk unitcell, and a $3 \times 3 \times 1$ one for monolayer calculations, respectively.

**Chern number and Berry curvature calculations**. The Chern number for one energy band can be calculated by the integral of the Berry curvature over the whole BZ

$$C_n = \frac{1}{2\pi} \int_{BZ} \Omega_n(\mathbf{k}) d^2\mathbf{k} \qquad (3)$$

The Berry curvature $\Omega_n(\mathbf{k})$ of the $n$th band can be determined as[29,56,57]

$$\Omega_n(\mathbf{k}) = -2\text{Im} \sum_{m \neq n} \frac{\psi_n(\mathbf{k})|v_x|\psi_m(\mathbf{k})\psi_m(\mathbf{k})|v_y|\psi_n(\mathbf{k})}{(\varepsilon_{n\mathbf{k}} - \varepsilon_{m\mathbf{k}})^2} \qquad (4)$$

where $\psi_n(\mathbf{k})$ and $\varepsilon_{n\mathbf{k}}$ are the spinor Bloch wave function and eigenvalue of the $n$th band at $\mathbf{k}$ point, and $v_{x(y)}$ is the velocity operator. The total Chern number[29] of a system is the sum of $C_n$ over all the occupied bands ($n$):

$$C = \sum_n C_n = \frac{1}{2\pi} \sum_n \int_{BZ} \Omega_n(\mathbf{k}) d^2\mathbf{k} \qquad (5)$$

**2D Monte Carlo simulations**. We have performed the Monte Carlo simulations based on the 2D Heisenberg model with uniaxial anisotropy, which is written as

$$H_M = -\sum_{\langle i,j \rangle} J\mathbf{m}_i \cdot \mathbf{m}_j - \sum_i D(m_i^z)^2 \qquad (6)$$

where $\mathbf{m}_i$ and $m_i^z$ represent the magnetic moment and its $z$ component on the $i$th moment, and $i$, $j$ confines the NN neighboring $\mathbf{m}_i$ and $\mathbf{m}_j$. The $J$ is the exchange integral ($J = -\Delta E/8m^2$, where $\Delta E = E_{FM} - E_{AFM}$) with the assumption of $|\mathbf{m}_i| = m$. $D$ is the reduced onsite MAE, and $D = -\text{MAE}/m^2$. A $100 \times 100$ supercell containing 10,000 local magnetic moments are adopted, and each simulation lasts $10^9$ loops to relaxation and another $10^9$ loops to collect the physical quantities. In each loop, one moment is rotated to a random direction.

## Data availability

The data that support the findings of this study are available from the corresponding author upon reasonable request.

## Code availability

The MC code used for simulations is available by written request to the corresponding authors.

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

## Acknowledgements

B.C., X.Z., and D.L. acknowledge the support from NSFC (Nos. 11574188). J.W. and B.H. acknowledge the support from Science Challenge Project (No. TZ2016003), NSFC (No. 11574024) and NSAF (No. U1930402). B.C. also acknowledges the support from Shandong Provincial Natural Science Foundation (No. ZR2019MA64) and the NSFC (No. 11404188). Computations were performed at Tianhe2-JK at CSRC. The authors thank Dr. Z. Liu (Tsinghua University), Dr. S. Yin and Dr. M. Zhao (Shandong University) for helpful discussions.

## Author contributions

B.C. and B.H. directed the project. B.C. and X.Z. calculated the results. B.C., J.W., D. L., S. X. and B.H. analyzed the results. B.C., J.W. and B.H. wrote the manuscript. All authors discussed the results and commented.

## Competing interests

The authors declare no competing interests.
