## [Peer Review File · Nature Communications]

Reviewers' Comments:

Reviewer #1:

Remarks to the Author:

The manuscript "Realization of Lieb lattice in Covalent-organic Frameworks with Tunable Ferromagnetism" describes a theoretical/computational study. The main point of the authors is that recent experiments, in which an organic layered material was synthesized, are in fact the first realization of the so-called Lieb lattice in nature. While other forms of this lattice have been observed in the past, they were all artificial materials obtained by e.g. photonic, cold-atom, and/or surface patterning techniques. In its ideal form, the Lieb lattice is predicted to exhibit interesting phenomena such as the topological Mott insulator, superconductivity, ferromagnetism, and fractional quantum Hall effect. The authors show with *ab initio* and tight-binding calculations how the experimentally synthesized structure maps onto a Lieb lattice, albeit not an ideal Lieb lattice. In addition, the authors can reproduce the experimentally found ferromagnetism and suggest a mechanism for tuning it via external pressure.

The article itself is well-written and easy to follow. The arguments of the authors are also clear and their conclusions are supported by their own calculations. While I enjoyed reading this manuscript, I think the main issue is one of novelty. In essence, the main point of the authors is that a previously found/synthesized experimental structure maps onto the Lieb lattice. The authors then also confirm the already known ferromagnetic state of the material. While it is true that the authors explain why the material synthesized with carbon (sp²C-COF) does show ferromagnetism and the one with nitrogen (sp²N-COF) does not (which is an interesting insight), I am missing that the paper provides steps that move the field beyond what has already been reported in the previous experimental paper. I understand, the authors elucidate the experiments and provide further insight, but to me that does not constitute the major importance I would want to see in Nature Communications. For example, the Lieb lattice is predicted to show interesting phenomena such as the topological Mott insulator, superconductivity, ferromagnetism, and fractional quantum Hall effect---the previous experimental paper reports the interesting case of ferromagnetism and the authors choose to pursue that further, making the paper less novel. Would the authors instead have focused on some topological properties, superconductivity, or the fractional quantum Hall effect, which were not previously reported, I would have considered the manuscript more novel.

Besides the aspect of novelty, I do have some concerns about the methodology used in their calculations:

1) The main set of calculations was performed with VASP, using the PBE exchange-correlation functional. However, a subset of calculations were performed with GAUSSIAN, using the B3LYP functional. It is very bad practice to switch either code or functional during a project, but the authors switch both. Why have not all calculations been performed at exactly the same level (code and exchange-correlation functional)?

2) There are no details at all about how the data in Fig. 4c is calculated.

3) Both structures are actually layered structures that are---as the authors mention---bound by van der Waals interactions. However, the chosen PBE functional (which I assume was used for the bulk calculations) does not capture van der Waals interactions, making the results for e.g. the pressure study unreliable.

The finding of the authors, i.e. the mapping of the synthesized structure onto the theoretical Lieb lattice, is a nice result and worth reporting in potentially another journal. But, at this point, I cannot recommend this manuscript for publication in Nature Communications, unless the authors can convincingly show advances that go beyond previously reported results.

Reviewer #2:

Remarks to the Author:

The authors present a theoretical study of the electronic states of single layers of two sp² covalent organic frameworks: sp²C-COF and sp²N-COF. A recent experimental study (ref. 26) has shown that upon iodine doping sp²C-COF exhibits a ferromagnetic phase. The goal of the present paper is to identify the origin of this magnetic phase. To this end the authors interpret the results of DFT calculations in terms of a modulated Lieb-Lattice model.

I have serious objections about this work.

1)

The discussion on the Lieb lattice model (pag. 2-3 of the main text and pag. 1-2 of the supplemental material) is too trivial to be reported: the diagonalization of a 3X3 matrix is an easy exercise for any undergraduate student. At the same time the results are not appropriately reported: in Fig. 1 the band structure obtained for different choices of the model parameters, t and E , are superimposed in the same graph. At a first glance it may appear that more than 3 bands may coexist, while this is obviously not the case.

2)

The interpretation of DFT results in terms of Lieb lattice model is not convincing. The DFT band structure (Fig. 2) exhibits in the region of interest 6 bands, 3 occupied and 3 unoccupied, while the Lieb lattice (ideal or modulated) hosts 3 bands only. The authors compare TB and DFT results by considering in the first case the full band structure (occupied and unoccupied) and only the valence band in the second case. Overall the TB and DFT band structures are drastically different: a single Dirac cone, possibly split by on-site energies differences, and a single flat band. In DFT one Dirac cone and one flat band in the conduction band, one split Dirac cone and one almost-flat band in the valence.

As a consequence the interpretation of the origin of ferromagnetism in the systems of interest here, based on this Lieb-lattice identification, is not justified.

3)

Moreover, ferromagnetism arises in the Lieb lattice as a consequence of many body effects associated to on site (Hubbard) e-e interaction, not in the single-particle picture, either DFT or TB. Indeed most of the interesting physical phenomena that are mentioned in the introduction (topological Mott insulator, superconductivity, ferromagnetism, fractional quantum (spin) Hall and the quantum anomalous Hall effects) as reported in references 3-9 of the manuscript, are due to many body effects.

For these reasons the main motivation of the paper – to identify sp²-COF as the first material realization of a Lieb-lattice – is not demonstrated and I do not recommend the paper for publication on Nature Communications.

The DFT calculation alone, even if rather standard, might be considered for publication in a more appropriate journal.

Here some minor comments:

a)Ref. 26 does not discuss sp²N-COF. Where have experimental data on sp²N-COF taken from?

b)Fig. 3 and the discussion of the effects of hole doping in terms of 1D chain model is rather confused and not fully convincing.

c)It is not clear how the Projected bulk band structure (fig. of supplementary material) is obtained. More details are necessary.

SUMMARY of REVISION

1. We have replied all the comments raised from both Referees and completely rewritten our manuscript and supplementary materials. The title of our manuscript is also updated.
2. We have extended the scope of our manuscript to include both the topological and magnetic properties of the organic-ligand-based Lieb lattice with distortions. All the figures in the Main Text are updated with new discussions.
3. The TB models are updated in the revised manuscript (Also see all the details in the supplementary materials). The DFT+ U method is performed to evaluate the effect of the onsite e-e interaction within the DFT methods, as shown in Fig. S8 in Page 8 in the Supplementary materials.
4. All the results calculated from the Gaussian code package from are removed.

Reply to Reviewer #1:

Question 1: *The manuscript “Realization of Lieb lattice in Covalent-organic Frameworks with Tunable Ferromagnetism” describes a theoretical/computational study. The main point of the authors is that recent experiments, in which an organic layered material was synthesized, are in fact the first realization of the so-called Lieb lattice in nature. While other forms of this lattice have been observed in the past, they were all artificial materials obtained by e.g. photonic, cold-atom, and/or surface patterning techniques. In its ideal form, the Lieb lattice is predicted to exhibit interesting phenomena such as the topological Mott insulator, superconductivity, ferromagnetism, and fractional quantum Hall effect. The authors show with ab initio and tight-binding calculations how the experimentally synthesized structure maps onto a Lieb lattice, albeit not an ideal Lieb lattice. In addition, the authors can reproduce the experimentally found ferromagnetism and suggest a mechanism for tuning it via external pressure.*

The article itself is well-written and easy to follow. The arguments of the authors are also clear and their conclusions are supported by their own calculations. While I enjoyed reading this manuscript, I think the main issue is one of novelty. In essence, the main point of the authors is that a previously found/synthesized experimental structure maps onto the Lieb lattice. The authors then also confirm the already known

ferromagnetic state of the material. While it is true that the authors explain why the material synthesized with carbon (sp²C-COF) does show ferromagnetism and the one with nitrogen (sp²N-COF) does not (which is an interesting insight), I am missing that the paper provides steps that move the field beyond what has already been reported in the previous experimental paper. I understand, the authors elucidate the experiments and provide further insight, but to me that does not constitute the major importance I would want to see in Nature Communications. For example, the Lieb lattice is predicted to show interesting phenomena such as the topological Mott insulator, superconductivity, ferromagnetism, and fractional quantum Hall effect---the previous experimental paper reports the interesting case of ferromagnetism and the authors choose to pursue that further, making the paper less novel. Would the authors instead have focused on some topological properties, superconductivity, or the fractional quantum Hall effect, which were not previously reported, I would have considered the manuscript more novel.

Reply 1: We thank the Referee for the constructive suggestions on improving the novelty of our MS.

The topological Mott insulator, superconductivity and fractional quantum Hall effects was predicted in (ideal) Lieb lattice models, mainly mediated by the flat bands. However, these interesting strongly-correlated physics need to be unveiled by performing higher-level many-body methods such as the dynamic mean field theory or exact diagonalization methods, which is beyond the scope of our current study (also beyond our expertise). In order to satisfy the Referee's comment, we have added extensive calculations on the topological properties of distorted Lieb lattice using both TB modelling and first-principles calculations. Interestingly, we find that 1) The lattice distortion can dramatically affect the electronic and topological properties of Dirac-flat bands in a Lieb lattice. And sp²C-COF and sp²N-COF, realized in the experiments, are two typical examples of distorted organic-ligand-based Lieb lattices; 2) Besides of the realization of FM phase, which is already known in the experiments, we find that the hole doping at different concentrations can also drive the organic COF lattice into multiple exotic states, i.e., half-metallic Dirac state, quantum spin Hall (QSH), quantum anomalous Hall (QAH) states.

Please see all the detailed changes in the revised MS and Supplementary Materials. In the following, we will briefly go through all the Figures in our revised MS:

Figure 1 | Three-band Lieb Lattice Models (a) Geometrical structure of a Lieb lattice, which contains one corner site 1 and two edge sites 2 and 3 in the unitcell (marked by the blue dashed-lines). The strength of lattice (geometrical) distortion is indicated by the change of structural angle θ away from 90° . See text for the definitions of other parameters. (b) Brillouin zone of a Lieb lattice. It notes that M and M1, equal in an ideal Lieb lattice, are inequivalent in a distorted Lieb lattice ($\theta \neq 90^\circ$ or $t_2 \neq t_3$). Projected band structures of an ideal Lieb lattice ($dE = 0$ and $\theta = 90^\circ$) (c) without and (d) with NNN hopping ($t_2 = t_3 = 0.2t_1$). Projected band structures of distorted Lieb lattices under three representative situations: (e) $t_2 = 2t_3 = 0.2t_1$, (f) $dE = 2t_1$ and (g) $dE = 2t_1$ and $t_2 = 2t_3 = 0.2t_1$.

In order to give a complete picture of the electronic structures of Lieb lattice as a function of distortions, in Figure 1 in the updated MS, we have systematically studied the evolutions of band structures of Lieb lattices with different kinds of distortions using TB modelling (See all the related details in page 2-3 in the updated MS). Fig. 1g is corresponding to the realistic case of organic COFs with distorted Lieb lattice, as synthesized in the experiments.

Figure 2 | Realization of Distorted Lieb Lattices in COFs. (a) Top views of (a) sp^2C -COF and (b) sp^2N -COF in a 2×2 supercell. The corner (TPPy) and edge (BCNB or BAIB) ligands are marked in the green-dashed squares and blue-dashed ellipses, respectively. Structures of end groups on the edge sites, i.e., crylonitrile for BCNB and aldimine for BAIB, are enlarged and shown in the insets. Projected band structures (without SOC effects) of monolayer (c) sp^2C -COF and (d) sp^2N -COF, where the green and blue colors indicate that the charge densities are contributed by corner (TPPy) and edge (BCNB or BAIB) sites, respectively, and the partial charge distributions in real space are also plotted as the insets of (c) and (d). SOC induced bandgap ($\times 100$) around Dirac point between VB2 and VB3 in sp^2C -COF is also plotted as inset in (c). TB fittings of DFT band structures based on distorted Lieb lattice model for these top three valence bands in (c) and (d) are also plotted as dashed-lines. For (c) sp^2C -COF: $t_1 = 0.1$ eV, $t_2 = 0.4t_1$, $t_3 = 0.45t_2$, $dE = 5.9t_1$, while for (d) sp^2N -COF: $t_1 = 0.12$ eV, $t_2 = 0.2t_1$, $t_3 = 0.45t_2$, $dE = 2.5t_1$.

In Figure 2, we have studied electronic structures of sp^2C -COF and sp^2N -COF based on the first-principles calculations. Our electronic structure analysis and TB band fitting both confirm that sp^2C -COF and sp^2N -COF are indeed the first material realizations of distorted organic-ligand-based Lieb lattices, whose electronic bands around the Fermi level can be well understood by the Dirac-flat bands models in Figures 1 (See all the related details in page 4-5 in the updated MS).

Figure 3 | Distorted Lieb Lattice Model with SOC Effects. Calculated band structures of distorted Lieb Lattices with SOC effects ($\lambda = 0.1t_1$). TB parameters of sp^2C -COF are selected for the modelling calculations, i.e., $t_1 = 0.1 \text{ eV}$, $t_2 = 0.4t_1$, $t_3 = 0.45t_2$, besides a variable dE : **(a)** $dE = 0 < dE_c$, **(b)** $dE = dE_c$, and **(c)** $dE = 5.9t_1 > dE_c$. $dE_c = 2\sqrt{(t_2 - t_3)^2 - 4\lambda^2}$. Calculated spin Chern number (C_s) for the Dirac-flat bands are also shown in these figures. Spin-resolved edge density of states for each band structure in **(a)**-**(c)** are also plotted in the right panels.

In order to give a complete physical picture of topological properties in distorted Lieb lattices, we further consider the SOC effects based on TB modelling, as shown in Figure 3 in the revised MS. Overall, we find that the topological phase transitions can occur for the Dirac-flat bands with changing dE (the onsite energy difference between the corner and edge sites in a distorted Lieb lattice), as indicated by the calculated spin Chern number (C_s) in Figure 3. The critical value of dE , i.e., dE_c , is found to be determined by the NNN hopping t_2 , t_3 , and SOC strength λ (See all the related details in page 6-8 in the updated MS). Correspondingly, the topological edge states appear connecting the bands with nonzero C_s . The topological properties of valence bands for sp^2C -COF or sp^2N -COF are corresponding to the case of Fig. 3c.

Figure 4 | Ferromagnetism in $sp^2\text{C-COF}$ and $sp^2\text{N-COF}$. DFT-calculated spin-polarized band structures of monolayer (a) $sp^2\text{C-COF}$ and (b) $sp^2\text{N-COF}$ with 0.5 holes doped in each unitcell. The calculated exchange energy splitting (E_s) is about 85 meV and 40 meV for $sp^2\text{C-COF}$ and $sp^2\text{N-COF}$, respectively. (c) Calculated magnetization as a function of temperature for monolayer $sp^2\text{C-COF}$ and $sp^2\text{N-COF}$ by Monto Carlo simulations. The transition temperatures (T_c) are marked with triangles.

In Figure 4 in the revised MS, we have calculated the electronic structures of FM phases in $sp^2\text{C-COF}$ and $sp^2\text{N-COF}$ after hole doping (0.5 hole/unitcell), corresponding to the iodine doping in the experiments. The different T_c values of $sp^2\text{C-COF}$ and $sp^2\text{N-COF}$ can be well understood by the different band widths of top valence band, which is mainly determined by the different dE and t_1 values in these two COFs (See all the related details in page 8-9 in the updated MS).

Figure 5 | Half-Metallic Dirac Cone and QAH Effects. (a) DFT-calculated spin-polarized band structures of monolayer sp^2C -COF with three hole doping per unitcell. E_F is set to zero. (b) Calculated SOC gap ($\times 100$) Δ_{24} in the spin down channel between VB2 and VB3 around E_F . (c) Spin-resolved edge density of states for (b) is also plotted in the right panel.

Finally, we discover that the half-metallic phase (without SOC effects) and QAH effect (with SOC effects) can be realized in sp^2C -COF when we further increase the hole doping concentrations to make VB2 partially-occupied with spontaneous spin-polarization (See all the related details in page 10-11 in the updated MS), which might be confirmed in the future experiments in these two COFs.

In summary, our new findings in the revised MS not only can explain the observed ferromagnetic (FM) ordering in the hole-doped sp^2C -COF in the experiments, but also suggests an interesting routine to realize the topologically nontrivial phases in the frustrated organic lattices. We wish the Referee can reevaluate our MS in terms of its sufficient novelty for the Nature Communications.

Question 2: Besides the aspect of novelty, I do have some concerns about the

methodology used in their calculations:

The main set of calculations was performed with VASP, using the PBE exchange-correlation functional. However, a subset of calculations were performed with GAUSSIAN, using the B3LYP functional. It is very bad practice to switch either code or functional during a project, but the authors switch both. Why have not all calculations been performed at exactly the same level (code and exchange-correlation functional)?

Reply 2: We thank the Referee's comment. In fact, the Gaussian calculations are not necessary in our updated MS. Therefore, we have removed all Gaussian calculations in our revised MS.

Question 3: *There are no details at all about how the data in Fig. 4c is calculated.*

Reply 3: We apologize for the mistake. The details of the MC simulations now can be found in the **Method** part in the MS:

The 2D Ising model Monte Carlo (MC) simulations are adopted by taking only the NN spin exchange energies into account. The 2D Ising Hamiltonian is written as $H_{\text{Is}} = -J \sum_{\langle i,j \rangle} m_i^z m_j^z$, where m_i^z and m_j^z are the local magnetic moments on the NN i th and j th sites, respectively, and the exchange integral J between them can be determined from $J = |\Delta E|/8m^2$, with assumption of $|m_i^z| = m$. A 100×100 supercell containing 10000 local magnetic moments are adopted, and each simulation lasts 10^9 loops. In each loop, each moment is changed to positive or negative m .

Question 4: *Both structures are actually layered structures that are---as the authors mention---bound by van der Waals interactions. However, the chosen PBE functional (which I assume was used for the bulk calculations) does not capture van der Waals interactions, making the results for e.g. the pressure study unreliable.*

Reply 4: Again, we apologize for our mistake. In fact, we have considered the van der Waals (vdW) corrections in the calculations of bulk COFs i.e., the DFT-D3 correction in the VASP package. We also want to mention that only the results of monolayer COFs are shown in the main text in the updated MS, and all the results on the bulk COFs with vdW corrections are moved to the Supplementary Materials (Fig.S2).

Question 5: *The finding of the authors, i.e. the mapping of the synthesized structure onto the theoretical Lieb lattice, is a nice result and worth reporting in potentially another journal. But, at this point, I cannot recommend this manuscript for publication in Nature Communications, unless the authors can convincingly show advances that go beyond previously reported results.*

Reply 5: As discussed in **Reply 1**, we have added sufficiently new results in our revised MS. (1) We conclude that the lattice distortions can dramatically affect the electronic and topological properties of Dirac-flat bands in a Lieb lattice, and COFs (realized in the experiments) are the first material realizations of distorted organic-ligand-based Lieb lattices; (2) Besides of the realization of FM phase, which is already known in the experiments, we further discover that hole doping at different concentrations can also drive the distorted organic-ligand-based COFs into multiple exotic states, i.e., half-metallic Dirac state, quantum spin Hall (QSH), quantum anomalous Hall (QAH) states.

We have completely rewritten the MS. We wish the Referee can reevaluate and support our MS publication in Nature Communications.

Reply to Reviewer #2:

Question 1: *The discussion on the Lieb lattice model (pag. 2-3 of the main text and pag. 1-2 of the supplemental material) is too trivial to be reported: the diagonalization of a 3X3 matrix is an easy exercise for any undergraduate student. At the same time the results are not appropriately reported: in Fig. 1 the band structure obtained for different choices of the model parameters, t and E , are superimposed in the same graph. At a first glance it may appear that more than 3 bands may coexist, while this is obviously not the case.*

Reply 1: We thank the Referee's comment. We have updated the TB modelling in our revision by considering more paramters. We have rewritten the entire MS based on both Referees' comments (please also see the **Reply 1 to Referee 1**), including all Figures.

For your convenience, we have copied the updated Figure 1 to here. In Figure 1 in the revised MS, we have systematically studied the evolutions of band structures of Lieb lattices with different distortions using TB modelling (See all the related details in page 2-4 in the updated MS). Fig. 1g is corresponding to the realistic case of organic COFs in the experiments with distorted Lieb lattice with both $\theta \neq 90^\circ$ and $dE \neq 0$.

Figure 1 | Three-band Lieb Lattice Models (a) Geometrical structure of a Lieb lattice, which contains one corner site 1 and two edge sites 2 and 3 in the unitcell (marked by the blue dashed-lines). The strength of lattice (geometrical) distortion is indicated by the change of structural angle θ away from 90° . See text for the definitions of other parameters. (b) Brillouin zone of a Lieb lattice. It notes that M and M1, equal in an ideal Lieb lattice, are inequivalent in a distorted Lieb lattice ($\theta \neq 90^\circ$ or $t_2 \neq t_3$). Projected band structures of an ideal Lieb lattice ($dE = 0$ and $\theta = 90^\circ$) (c) without and (d) with NNN hopping ($t_2 = t_3 = 0.2t_1$). Projected band structures of distorted Lieb lattices under three representative situations: (e) $t_2 = 2t_3 = 0.2t_1$, (f) $dE = 2t_1$ and (g) $dE = 2t_1$ and $t_2 = 2t_3 = 0.2t_1$.

Question 2: *The interpretation of DFT results in terms of Lieb lattice model is not convincing. The DFT band structure (Fig. 2) exhibits in the region of interest 6 bands, 3 occupied and 3 unoccupied, while the Lieb lattice (ideal or modulated) hosts 3 bands only. The authors compare TB and DFT results by considering in the first case the full band structure (occupied and unoccupied) and only the valence band in the second case. Overall the TB and DFT band structures are drastically different: a single Dirac cone, possibly split by on-site energies differences, and a single flat band. In DFT one Dirac cone and one flat band in the conduction band, one split Dirac cone and one almost-flat band in the valence.*

As a consequence the interpretation of the origin of ferromagnetism in the systems of interest here, based on this Lieb-lattice identification, is not justified.

Reply 2: We thank the Referee for his/her constructive comments. We have replotted the band structures of sp^2 C-COF and sp^2 N-COF by including the M1 point in the BZ (M and M1, marked in Figure 1, are not equal in a distorted Lieb lattice), which is missed in our initial band structure plotting.

Now, we can perfectly fit both the top three valence bands (for sp^2 C-COF: $t_1 = 0.1 \text{ eV}, t_2 = 0.4t_1, t_3 = 0.45t_2, dE = 5.9t_1$, for sp^2 N-COF: $t_1 = 0.12 \text{ eV}, t_2 = 0.2t_1, t_3 = 0.45t_2, dE = 2.5t_1$.) and the bottom three conduction bands (for sp^2 C-COF: $t_1 = 0.12 \text{ eV}, t_2 = -0.1t_1, t_3 = 0.45t_2, dE = 0.4t_1$, for sp^2 N-COF: $t_1 = 0.15 \text{ eV}, t_2 = -0.1t_1, t_3 = 0.45t_2, dE = -0.4t_1$) using distorted Lieb lattice TB models. Due to the different wavefunction characters of top valence bands (contributed by HOMO of TPPy) and bottom conduction bands (contributed by LUMO of BCNB or BAIB), the TB band fitting parameters are different. The reason that we only focus on the the top three valence bands in our initial submission is because only the top valence bands can play a critical role in determining the FM phase of sp^2 C-COF and sp^2 N-COF after ionic oxidation (hole doping).

To response the Referee's comment, we have shown the TB band fitting of bottom three conduction bands of sp^2 C-COF and sp^2 N-COF in Fig. R1 in the following. For your convenience, we have copied the updated Figure 2 to here (See all the related details in page 8 in the updated MS). Fig. R1 is also put in the Supplementary Materials as Fig. S4.

Figure R1 | TB Fitting of Bottom Three Conduction Bands in COFs TB band fitting (orange-dashed lines) for the three bottom conduction bands of (a) sp^2 C-COF and (b) sp^2 N-COF. The relative energy levels of the frontier molecular orbitals for the ligands are depicted in the insets. For (a) sp^2 C-COF: $t_1 = 0.12 \text{ eV}, t_2 = -0.1t_1, t_3 = 0.45t_2, dE = 0.4t_1$, while for (b) sp^2 N-COF: $t_1 = 0.15 \text{ eV}, t_2 = -0.1t_1, t_3 = 0.45t_2, dE = -0.4t_1$

Figure 2 | Realization of Distorted Lieb Lattices in COFs. (a) Top views of (a) sp^2C -COF and (b) sp^2N -COF in a 2×2 supercell. The corner (TPPy) and edge (BCNB or BAIB) ligands are marked in the green-dashed squares and blue-dashed ellipses, respectively. Structures of end groups on the edge sites, i.e., crylonitrile for BCNB and aldimine for BAIB, are enlarged and shown in the insets. Projected band structures (without SOC effects) of monolayer (c) sp^2C -COF and (d) sp^2N -COF, where the green and blue colors indicate that the charge densities are contributed by corner (TPPy) and edge (BCNB or BAIB) sites, respectively, and the partial charge distributions in real space are also plotted as the insets of (c) and (d). SOC induced bandgap ($\times 100$) around Dirac point between VB2 and VB3 in sp^2C -COF is also plotted as inset in (c). TB fittings of DFT band structures based on distorted Lieb lattice model for these top three valence bands in (c) and (d) are also plotted as dashed-lines. For (c) sp^2C -COF: $t_1 = 0.1$ eV, $t_2 = 0.4t_1$, $t_3 = 0.45t_2$, $dE = 5.9t_1$, while for (d) sp^2N -COF: $t_1 = 0.12$ eV, $t_2 = 0.2t_1$, $t_3 = 0.45t_2$, $dE = 2.5t_1$.

Question 3: Moreover, ferromagnetism arises in the Lieb lattice as a consequence of many body effects associated to on site (Hubbard) e - e interaction, not in the

single-particle picture, either DFT or TB. Indeed most of the interesting physical phenomena that are mentioned in the introduction (topological Mott insulator, superconductivity, ferromagnetism, fractional quantum (spin) Hall and the quantum anomalous Hall effects) as reported in references 3-9 of the manuscript, are due to many body effects.

Reply 3: We agree with the Referee that ferromagnetism originates from the many-body effects, which is usually underestimated by the mean-field approximation within the DFT methods. The accurate calculations on strongly-correlated systems need to be unveiled by performing higher-level many-body methods such as the dynamic mean field theory or exact diagonalization methods, which is indeed far beyond the scope of our current study (the main scope of our study is to demonstrate the first material realization of Lieb lattices in organic COFs).

In fact, the FM ground states of enormous systems (some of them are similar to our system) have been correctly evaluated (as shown in Table R1 in the following). Especially, the onsite $e-e$ interactions in the sp^2 C-COF and sp^2 N-COF (sp^2 -hybridized C and N systems) is very weak compared to the traditional magnetic ($3d$ - or $4f$ -) systems. On the other hand, the widely acceptable remedy within the DFT formalism to include the electron correlations is the Hartree–Fock mean-field method, i.e., DFT+ U correction. We have performed a GGA+ U calculation with $U=1.5$ eV on the $2p$ orbitals of C and N atoms, and our calculations show that the enhanced U can increase the exchange splitting (E_s), as shown in Fig. R2 [also Fig. S8 in the Supplementary Materials], and hence can further enhance the stability of FM ground-states. Therefore, the conclusion of FM ordering in hole-doped COFs obtained from DFT calculations is still valid, which is also consistent with the experimental measurements.

Figure R2 | DFT+ U calculated band structure for sp^2 C-COF with $U = 0$ and $U = 1.5$ eV.

Table R1: Examples of DFT (or DFT+ U) calculations for magnetic ground-states:

Articles	Authors	Method	Conclusion
----------	---------	--------	------------

“Half-metallic graphene nanoribbons.” Nature 444, 5-8(2006) [cited by 3592 times]; “Energy Gaps in Graphene Nanoribbons.” Phys. Rev. Lett. 97, 16803 (2006) [cited by 4193 times]	Louie, S. G. et al.	DFT	DFT and TB calculations predict the magnetic ground-states of zigzag graphene nanoribbons with flat band (edge states). The theoretical predictions have been confirmed in the later experiments [Nature 531, 7595 (2016), Nature 514, 7524, (2014)].
“Two-dimensional materials from high-throughput computational exfoliation of experimentally known compounds.” Nat. Nanotechnol 13, 246–252 (2018) [cited by 128 times].	Mazari, N. et al.	DFT	This work performed large-scale DFT calculations and identified 56 FM and AFM systems from the ICSD database (some of them have already confirmed by the experiments).
“Origin and Enhancement of Hole-Induced Ferromagnetism in First-Row d^0 Semiconductors”. Phys. Rev. Lett. 102, 017201 (2009) [cited by 340 times].	Wei, S.-H. et al.	DFT	These two works explain the hole-doping-induced ferromagnetism in d^0 semiconductors, e.g., ZnO. The origin of hole-doping-induced ferromagnetism in their systems is similar to our study, which can
“Room-Temperature Ferromagnetism in Carbon-Doped ZnO.” Phys. Rev. Lett. 99, 127201 (2007) [cited by 723 times].	Ding, J. et al.	DFT	be understood by the Stoner’s criteria.
“Tunable magnetism and half-metallicity in hole-doped monolayer GaSe” Phys. Rev. Lett. 114, 236602 (2015) [cited by 145 times].	Louie, S. G. et al.	DFT	They predict the existence of FM ground-state with hole doping in the top valence band in 2D GaSe. The origin of hole-doping-induced ferromagnetism in their systems is similar to our study, which can be understood by the Stoner’s criteria.

“Ferromagnetism in GaN:Gd: A density functional theory study.” *Phys. Rev. Lett.* **100**, 127203 (2008) [cited by 141 times].

Mao, S. S.
et al.

DFT+U

The experimental colossal magnetic moments and room-temperature ferromagnetism in GaN:Gd are explained by the interaction of Gd 4f spins via p-d coupling involving holes introduced by intrinsic defects such as Ga vacancies.

Finally, we want to emphasize that we have added the extensive calculations on the topological properties of distorted Lieb lattices in our revision based on Referee 1’s comments, as shown in Figure 3 and Figure 5 (**Please see our Reply 1 to Referee 1 and also the revised MS**).

Question 4: Minor comment a): *Ref. 26 does not discuss sp^2N -COF. Where have experimental data on sp^2N -COF taken from?*

Reply 4: In the supporting information of the experimental paper, the authors show the weak ESR signal in Fig. S12. The geometrical structures of sp^2N -COF and sp^2C -COF are similar. In our calculations, the fully geometrical relaxations are carried out, as shown in the **Method part** in the MS.

Question 5: Minor comment b): *Fig. 3 and the discussion of the effects of hole doping in terms of 1D chain model is rather confused and not fully convincing.*

Reply 5: Thank you for this comment, and we have deleted this figure in the revision, as it is no longer needed. We also want to mention that we have updated all the figures in the revised MS.

Question 6: Minor comment c): *It is not clear how the Projected bulk band structure (fig. of supplementary material) is obtained. More details are necessary.*

Reply 6: The bulk band structures (Fig. S2 in the Supporting Materials) are calculated with the consideration of vdW interactions using DFT-D3 correction, as implemented in VASP code.

For the band projection:

- 1) The eigenstate can be written as a superposition of atomic orbital (AO) wavefunctions: $\Phi_{\mu,\mathbf{k}}(\mathbf{r}) = \sum_i a_i(\mathbf{k}, \mathbf{r}) \phi_i^l(\mathbf{r})$, where ϕ_i^l is the l th atomic orbital on the i th atom, e.g., p_i^z (the p_z orbital on the i th atoms).
- 2) Then, the weight of ϕ_i^l in the $\Phi_{\mu,\mathbf{k}}(\mathbf{r})$ is able to be obtained from the

projection:

$$a_i(\mathbf{k}, \mathbf{r}) = \langle \phi_i^l | \Phi_{\mu, \mathbf{k}}(\mathbf{r}) \rangle$$

3) The contribution from the ϕ_i^l orbitals on the two ligands are

$$A_j(\mathbf{k}) = \sum_{i \in \{j\}} a_i(\mathbf{k}) = \sum_{i \in \{j\}} \langle \phi_i^l | \Phi_{\mu, \mathbf{k}}(\mathbf{r}) \rangle, \quad j = 1, 2$$

where $\{1\}$ ($\{2\}$) denotes the atom subset on corner (edge) ligand marked by the green (blue) colors in Figs. 2c and 2d in the updated SM.

For the partial charge density distribution:

The partial charge density distributions of the VB1-VB3 are obtained by the summations of the inner products of the eigenstates of the corresponding bands:

$$\rho_{\mu}(\mathbf{r}) = \langle \Phi_{\mu, \mathbf{k}}(\mathbf{r}) | \Phi_{\mu, \mathbf{k}}(\mathbf{r}) \rangle_{\mathbf{k}}.$$

Reviewers' Comments:

Reviewer #1:

Remarks to the Author:

I have read all the materials related to the resubmitted manuscript "Realization of Lieb Lattice in Covalent-organic Frameworks with Tunable Topology and Ferromagnetism". In my original report, I have brought up as my main point one of novelty, as the original paper merely elucidates on previously published experimental results. Besides that, I have brought up 3 minor points. All minor points have been nicely addressed by the authors and with regards to those I have no concerns left. With respect to the main point of novelty, I acknowledge that the authors have reorganized and rewritten major parts of the manuscript, including interesting aspects that add to the novelty. The main addition now investigates the topological nature of the various Lieb lattice bands and a short section also discusses the possibility of finding the quantum anomalous Hall effect in those materials. The topological aspects are assessed in detail and a thorough discussion of spin-orbit effects is included. I think that at this point the manuscript could be interesting to the large audience in topological insulators, which is a field of much current interest. I thus can now recommend publication of the manuscript in its current form.

Reviewer #2:

Remarks to the Author:

I am sorry to say that this revised version of the manuscript is even less convincing than the previous ones.

The proposed interpretation of the DFT results in terms of Tight-Binding parametrization (and consequently of Lieb Lattice physics) that is at the basis of the study is simply wrong.

The DFT band structure in fact exhibits in the region of interest 6 bands, 3 occupied and 3 unoccupied, while the Lieb lattice -ideal or modulated- hosts 3 bands only. As it is well known by elementary solid state theory, the number of bands that are obtained in a Tight-Binding scheme is given by $N_s \times N_o$, where N_s is the number of inequivalent sites in the unit cell number and N_o the number of orbitals per site. The authors seem to ignore this basic notion and simply "fit" separately the valence and conduction bands with different hoppings. This is meaningless. A correct Tight-Binding parametrization, that I stress again is nothing else than a change of basis, should provide an unique set of parameters that reproduce both valence and conduction bands, the gap width and the position of the Fermi level.

For this reason all the analysis based on the proposed Tight-Binding parametrization is pointless.

Moreover, a new discussion on topological properties is proposed, based on the introduction of a complex hopping in the Kane-Mele scheme to describe spin-orbit interaction. The authors calculate the band spin Chern numbers and from them they attribute a topological character to each band. This is a severe misinterpretation: topological properties are in fact associated to the TOTAL Chern number, namely to the sum of the band Chern numbers over the occupied states. The total Chern number – and not the band Chern number, is associated to conductivity and therefore topologically protected edge states are expected only when the total Chern number is non zero.

The discussion of the topological character of the Lieb lattice alone is therefore invalidated by this misinterpretation. Even more so for the extension to the real material where the valence band is fully occupied.

In conclusion I think that the paper contains serious basic mistakes and should not be published neither in Nature Communications nor in any other Journal.

Summary of changes

1. Based on Reviewer 2's comment, we have rewritten the TB band fittings based on a 6-orbital model. The fitting results are shown in Fig. S5 in the Supplementary Materials.
2. Based on Reviewer 2's comment, we have switched the order of Fig. 2 and Fig. 3 and made some related changes on the discussion of (spin) Chern number calculations.

Reply to Reviewer #1:

Comment: I have read all the materials related to the resubmitted manuscript "Realization of Lieb Lattice in Covalent-organic Frameworks with Tunable Topology and Ferromagnetism". In my original report, I have brought up as my main point one of novelty, as the original paper merely elucidates on previously published experimental results. Besides that, I have brought up 3 minor points. All minor points have been nicely addressed by the authors and with regards to those I have no concerns left. With respect to the main point of novelty, I acknowledge that the authors have reorganized and rewritten major parts of the manuscript, including interesting aspects that add to the novelty. The main addition now investigates the topological nature of the various Lieb lattice bands and a short section also discusses the possibility of finding the quantum anomalous Hall effect in those materials. The topological aspects are assessed in detail and a thorough discussion of spin-orbit effects is included. I think that at this point the manuscript could be interesting to the large audience in topological insulators, which is a field of much current interest. I thus can now recommend publication of the manuscript in its current form.

Reply: We thank Reviewer 1 for the recommendation of publication of our manuscript in the present form.

Reply to Reviewer #2:

Question 1: The proposed interpretation of the DFT results in terms of Tight-Binding parametrization (and consequently of Lieb Lattice physics) that is at the basis of the study is simply wrong. The DFT band structure in fact exhibits in the region of interest 6 bands, 3 occupied and 3 unoccupied, while the Lieb lattice -ideal or modulated- hosts 3 bands only. As it is well known by elementary solid state theory, the number of bands that are obtained in a Tight-Binding scheme is given by $N_s \times N_o$, where N_s is the

number of inequivalent sites in the unit cell number and N_o the number of orbitals per site. The authors seem to ignore this basic notion and simply “fit” separately the valence and conduction bands with different hoppings. This is meaningless. A correct Tight-Binding parametrization, that I stress again is nothing else than a change of basis, should provide an unique set of parameters that reproduce both valence and conduction bands, the gap width and the position of the Fermi level. For this reason all the analysis based on the proposed Tight-Binding parametrization is pointless.

Reply 1: According to the Lowdin perturbation theory, it is very universal to project the full Hamiltonian to a subspace using the downfolding technique. For example, in Walter Harrison’s classical textbook “*Electronic Structure and the Properties of Solids*”, the band edge of Si can be **either** described by an 8×8 matrix using the sp^3 atomic orbitals as the basis **or** by two 4×4 matrices using the hybridized bonding and antibonding orbitals as the basis, respectively.

In our systems, we used the three-band Lieb lattice model to separately fit the top three valence bands (TTVBs) and bottom three conduction bands (BTCBs), because the molecular orbitals (MOs) of TTVBs and BTCBs are contributed by the HOMOs and LUMOs of ligands, respectively. In this situation, the tight-binding (TB) fitting parameters of TTVBs and BTCBs are different, as they have different orbital basis. Again, **we want to emphasize that this is very common that the conduction and valence bands are fitted by different TB parameters, which is widely reported in the literature.** For example, Brédas *et al.* [J. Brédas *et al.*, PNAS **99**, 5804–5809 (2002)] have demonstrated that the TB fitting parameters of conduction and valence bands are usually different in a large range of n-thiophene- and n-acene-based organic molecular crystals; Scriven and Powell [E. P. Scriven and B. J. Powell, Phys. Rev. Lett. **109**, 097206 (2012)] have used a similar way as ours to fit the conduction and valence bands of $\text{Me}_3\text{EtP}[\text{Pd}(\text{dmit})_2]_2$ and $\text{Me}_3\text{EtSb}[\text{Pd}(\text{dmit})_2]_2$ organic crystalline systems separately; Z. Liu *et al.* [Z. Liu *et al.* Phys. Rev. Lett. **110**, 106804 (2013)] have also selectively fitted several valence bands, instead of fitting both the conduction and valence bands, to evaluate the Chern flat band in the indium-phenylene organometallic framework.

In order to do our best to satisfy the Reviewer’s concern, we have further performed a 6-orbital TB fitting, including both TTVBs and BTCBs in both $sp^2\text{C-COF}$ and $sp^2\text{N-COF}$. The results are shown in Fig. R1.

Figure. R1 | (a) and (c) 6-orbital TB band fitting (orange-dashed lines) of the top three valence bands and bottom three conduction bands for the DFT-calculated sp^2C -COF and sp^2N -COF band structures, respectively. (b) and (d) Corresponding TB band structures with band projections for the sp^2C -COF and sp^2N -COF, respectively. Blue and green colors denote the contributions of the BCNB (BAIB) and TPPy in sp^2C -COF (sp^2N -COF), respectively. For sp^2C -COF: $t_{1H} = t_1 = 0.1\text{eV}$, $t_{2H} = 0.4t_1 = 0.04\text{eV}$, $t_{3H} = 0.45t_{2H} = 0.018\text{eV}$, $dE_H = 5.9t_1$, $\varepsilon_{1H} = -0.135\text{eV}$, $\varepsilon_{2H} = \varepsilon_{3H} = -0.725\text{eV}$, $t_{1L} = 1.2t_{1H} = 0.12\text{eV}$, $t_{2L} = -0.1t_{1L} = -0.04\text{eV}$, $t_{3L} = 0.45t_{2L} = -0.018\text{eV}$, $dE_L = 0.4t_{1L}$, $\varepsilon_{1H} = 1.79\text{eV}$, and $\varepsilon_{2H} = \varepsilon_{3H} = 1.75\text{eV}$. For sp^2N -COF: $t_{1H} = t_1 = 0.12\text{eV}$, $t_{2H} = 0.4t_1 = 0.048\text{eV}$, $t_{3H} = 0.45t_{2H} = 0.022\text{eV}$, $dE_H = 2.5t_1$, $\varepsilon_{1H} = -0.222\text{eV}$, $\varepsilon_{2H} = \varepsilon_{3H} = -0.522\text{eV}$, $t_{1L} = 1.25t_{1H} = 0.15\text{eV}$, $t_{2L} = -0.1t_{1L} = -0.015\text{eV}$, $t_{3L} = 0.45t_{2L} = -0.007\text{eV}$, $dE_L = 0.4t_{1L}$, $\varepsilon_{1H} = 1.686\text{eV}$, and $\varepsilon_{2H} = \varepsilon_{3H} = 1.746\text{eV}$. Insets of (a) and (c): Band alignment for molecular orbitals (HOMOs and LUMOs) of the ligands.

The 6-orbital hamiltonian is given as:

$$H = \begin{pmatrix} H_H & H_{HL} \\ H_{HL}^\dagger & H_L \end{pmatrix},$$

where the diagonal blocks are 3-orbital hamiltonian for the HOMOs (H_H) and LUMOs (H_L). It can be written as:

$$H_{H/L} = \begin{pmatrix} E_1 & E_{12} & E_{13} \\ E_{21} & E_2 & E_{32} \\ E_{31} & E_{32} & E_3 \end{pmatrix}.$$

$E_1 = \varepsilon_{1\text{H/L}}$ and $E_2 = E_3 = \varepsilon_{2\text{H/L}}$. The NN hoppings are $E_{12} = -2t_{1\text{H/L}} \cos(k_1 a/2)$,
 $E_{21} = E_{12}^*$, $E_{13} = -2t_{1\text{H/L}} \cos(k_2 a/2)$ and $E_{31} = E_{13}^*$. The NNN hoppings are
 $E_{23} = -2t_{2\text{H/L}} \cos[(-k_1 + k_2) a/2] - 2t_{3\text{H/L}} \cos[(k_1 + k_2) a/2]$ and $E_{32} = E_{23}^*$.
 H_{HL} and H_{HL}^\dagger are the interactions between the HOMOs and LUMOs. Due to the large
band gaps of $sp^2\text{C-COF}$ and $sp^2\text{N-COF}$, H_{HL} and H_{HL}^\dagger are neglected. Therefore, the
6-orbital TB band fitting has similar features as our initial two 3-orbital TB band fitting
of TTVBs and BTCBs.

Based on the Reviewer's comment, we have added Fig. R1 and related discussions in
the Supplementary Materials as Fig. S5. In our main manuscript, we focus on the
discussion of TB band fitting of TTVBs, as only TTVBs play a critical role in the hole-
doping-induced FM and topological phases in these COFs.

Question 2: Moreover, a new discussion on topological properties is proposed, based
on the introduction of a complex hopping in the Kane-Mele scheme to describe spin-
orbit interaction. The authors calculate the band spin Chern numbers and from them they
attribute a topological character to each band. This is a severe misinterpretation:
topological properties are in fact associated to the TOTAL Chern number, namely to
the sum of the band Chern numbers over the occupied states. The total Chern number
– and not the band Chern number, is associated to conductivity and therefore
topologically protected edge states are expected only when the total Chern number is
non zero.

The discussion of the topological character of the Lieb lattice alone is therefore
invalidated by this misinterpretation. Even more so for the extension to the real material
where the valence band is fully occupied.

Reply 2: We agree with the Reviewer that only total (spin) Chern number is associated
with the observable quantum conductance of the system. However, **we want to
emphasize that the original definition of Berry curvature is based on the energy
band. Therefore, the (spin) Chern number can be unambiguously assigned to each
band** [e.g., see Fukui *et al.*, JPSJ **74**, 1674 (2005)]. A similar band-by-band analysis of
the kagome lattice led to the pioneering proposal of fractional Chern insulator [Phys.
Rev. Lett. **106**, 236802 (2011)]. More related work can refer to Phys. Rev. Lett. **105**,
017401 (2010); Phys. Rev. Lett. **109**, 186805 (2012); Phys. Rev. Lett. **110**, 116802
(2013); Phys. Rev. Lett. **115**, 087003 (2015); Nat. Commun. **5**, 5782 (2014). **There is
no wonder that the Hall conductance is a summation of the Chern numbers of all
the occupied bands, but it is also important to understand how each band
contributes to the sum.** For our case, although the total spin Chern number is zero

($C_s=0$) for both COF systems due to the fully occupied VB1-VB3, the nontrivial topology of Dirac-flat bands (reflected by spin Chern number calculations of each band) inspires us to study the electronic properties of these COFs under hole doping.

Figure R2 | Half-Metallic Semimetal and QAHE. (a) DFT-calculated spin-polarized band structure of monolayer sp^2C -COF with three holes doping into per unitcell. E_F is set to zero. (b) Calculated SOC gap ($\times 100$) $\Delta_{2\downarrow}$ in the spin down channel between VB2 and VB3 around E_F . The calculated Chern number, $C = -1$, is also marked in this figure. (c) Spin-resolved edge states for (b).

In Fig. 5 (copy here as Fig. R2) in our revised manuscript, we have calculated the total Chern number $C = -1$ for sp^2C -COF under a hole doping concentration of $\sim 10^{13} \text{ cm}^{-2}$, and such a carrier doping concentration may be accessible in the experiments [e.g., see D. K. Efetov and P. Kim, Phys. Rev. Lett. **105**, 246805 (2010); J. Ye *et al.* Science **338**, 1193 (2012)]. This nonzero Chern number demonstrates a quantum anomalous Hall effect (QAHE). The topological protected edge states are also be calculated, as shown in Fig. R2c (Please see Pages 9-10 in the revised manuscript for the details).

To make our MS more readable and smooth, we have switched the order of Fig. 2 and Fig. 3 and modified related discussions (Please see Pages 4-7 in the revised manuscript for the details). Especially, based on the Reviewer's comment, **we have defined C_n^S for the spin Chern number of n th band and highlighted that only the total (spin) Chern number C_s is associated to the observable conductivity in the related discussions in our revised manuscript.** For example, as shown in Page 5 of the revised

manuscript, we have added the sentences of **“It is emphasized that in a realistic material, only the total spin Chern number C_s , the sum of C_n^s for all the occupied bands, is associated with the observable quantum conductance of the system. Therefore, the topological properties of a realistic Lieb lattice material depend on the position of Fermi level, and charge doping may be needed to achieve a nontrivial state.”**

In summary, we wish the Reviewer 2 will be able to reevaluate our revised manuscript in terms of its novelty, significance, and validity, in consideration of the improvements we have made.

Reviewers' Comments:

Reviewer #3:

Remarks to the Author:

I am writing this report on the manuscript of Cui and co-workers ("Realization of Lieb Lattice in Covalent-organic Frameworks with Tunable Topology and Ferromagnetism") as additional referee (I did not review the first version), since the previous round of reviews ended up with a split decision. My review is not arbitrating between the authors and the previous referees, but it offers an independent view on the manuscript.

I have a number of serious concerns about this paper. The main one has to do with its novelty. A few months ago I have reviewed another paper for Nature Communication on the same topic titled "A Lieb-Like Lattice in a Covalent-Organic Framework and Its Unconventional Ferromagnetism". The two works are extremely similar to each other, namely they consider an identical tight-binding model, which is fitted onto DFT calculations for the same covalent-organic framework. As such they both reach the same conclusion concerning the possible magnetism in this system. I do not know what was the dynamics of the submission (who submitted first, how much time elapsed between the two submission, etc.) and I leave to the editorial team the decision on what to do. It is a fact, though, that the two papers present almost identical physics, so I cannot see the point to publish them both.

Having said that, I have a few technical concerns.

1) I believe that the analysis of the ferromagnetism is weak and largely incorrect. The authors find band spin-splitting in the metallic phase, as expected by the Stoner criterion applied to a narrow band related to a 2p shell. This is expected and reasonable. Their analysis of the critical temperature, however, is simply manufactured to get the right answer. Firstly, they map the DFT results on an Ising model. This has no ground, since the system does not possess uniaxial anisotropy (they did not calculate that, and anyway it is expected to be tiny in these organic compounds). The more appropriate mapping would be onto a Heisenberg model, which, of course, includes transverse spin fluctuations. Here the second problem arises. In 2D the Heisenberg model does not have a ferromagnetic phase transition if the coupling is short range, namely $T_C=0$. This is in virtue of the Wagner-Mermin theorem. In a nutshell, the authors map the (probably correct) band structure on the wrong model, just to have the experimental T_C , since a correct mapping would not give any T_C .

Note, in passing, that the T_C for the Ising model does not require any Monte Carlo simulations. It is a known analytical result, $T_C \cdot k_B = 2 / \ln(1 + \sqrt{2}) \cdot J$.

2) It seems to me that there is no relation between the spin-orbit coupling and the ferromagnetism. Also it is not clear whether or not SO is necessary for the half-metallic state. I don't think this is the case.

3) The authors should make an estimate (by DFT for instance) of the magnitude of the SO coupling. Is this practically relevant?

4) The hole concentrations investigated appear to me unrealistically large. It is true that ionic liquid gating can achieve such doping level, but this is usually done on insulators presenting a moderately large gap, which is not the case here.

5) There are a number of statements, which are simply incorrect. For instance: "The half-metallic gap is estimated to be as large as 0.1 eV in sp²C-COF, which is sufficiently large for high-temperature spintronic applications."

One cannot have room-temperature applications if the T_C is around 10K !!

In conclusion I believe that the present paper has a few technical issues that prevent its publication. In addition there is a major concern related to the similarity with another work. As such I cannot recommend publication.

Additional Note:

I have now taken a look at the rebuttal of the authors to the first round of review. Reviewer 1 is satisfied by the reply, so that only Reviewer 2 remains critical. Here is my view:

1) I think the reviewer is incorrect concerning the question of the 3-band model against the 6-band one (Question 1). One can always construct an effective model with a limited number of degrees of freedom, namely two 3-band model describing respectively the conduction and valence band (this is what was done by the authors in the reply to the referee's comment). Such approach fails if there is some non-trivial interaction between the conduction and the valence band (e.g. interaction between the two manifold). This does not seem to be the case here (the authors do not treat that case).

2) I believe that the referee is correct to ask Question 2. The reply from the authors seems compelling to me.

Reply to Reviewer #3:

Comment 1: I am writing this report on the manuscript of Cui and co-workers ("Realization of Lieb Lattice in Covalent-organic Frameworks with Tunable Topology and Ferromagnetism") as additional referee (I did not review the first version), since the previous round of reviews ended up with a split decision. My review is not arbitrating between the authors and the previous referees, but it offers an independent view on the manuscript.

I have a number of serious concerns about this paper. The main one has to do with its novelty. A few months ago I have reviewed another paper for Nature Communication on the same topic titled "A Lieb-Like Lattice in a Covalent-Organic Framework and Its Unconventional Ferromagnetism". The two works are extremely similar to each other, namely they consider an identical tight-binding model, which is fitted onto DFT calculations for the same covalent-organic framework. As such they both reach the same conclusion concerning the possible magnetism in this system. I do not know what was the dynamics of the submission (who submitted first, how much time elapsed between the two submission, etc.) and I leave to the editorial team the decision on what to do. It is a fact, though, that the two papers present almost identical physics, so I cannot see the point to publish them both.

Reply 1: We thank the Reviewer for this comment, and we understand this concern.

Firstly, we want to emphasize that our study is totally independent of that of Jiang *et al.* [*Nat. Commun.* **10**, 2207 (2019)], and we submitted our work when Jiang *et al.*'s paper wasn't published. Hence it is not fair to use Jiang *et al.*'s paper to argue against the novelty of our work. Since our manuscript is experiencing a very long-time review process (June 29th, 2018-now), its publication is significantly delayed.

Secondly, although our work shares several similarities with that of Jiang *et al.*, such as the discovery of first material realizations of Lieb lattice in COFs and the explanation of the possible ferromagnetism in these COFs after hole doping, we also report several interesting findings beyond Jiang *et al.* during the extensive revision, such as the antiferromagnetic ground state in the overdoping range (See Figure 5 in the main text). Here, we would like to list the key new findings below (beyond Jiang *et al.*):

1. We have studied the mechanism of the existence of a flatter top valence band in sp^2 C-COF than in sp^2 N-COF (see discussions in Page 7 in the main text), which is not only critical to explain the higher T_c in sp^2 C-COF, but also suggest a possible way to design flat bands in organic lattices. It is known that flat bands could have great importance for realizing many novel physical phenomena, e.g., the Mott insulating phase, ferromagnetism, and superconductivity in twist graphene systems are mainly induced by the flat bands around the Fermi level.

2. We have systematically studied the topological properties of distorted Lieb lattice models (see Figure 2 in the main text).

3. We have systematically calculated the magnetic phase transitions of sp^2 C-COF as a function of hole doping concentrations. We have discovered a paramagnetic-ferromagnetic-antiferromagnetic (PM-FM-AFM) phase transition as the hole concentration increases (see Figure 5 in the main text or Reply 5 in the following). This finding is very different from the prediction by Jiang *et al.*, who state that FM phase could be enhanced as hole doping concentration increases. Such PM-FM-AFM phase transition indicates that these COFs could serve as a suitable platform to study the correlation effects of electrons due to their flat bands.

In summary, we wish the Reviewer could re-evaluate our manuscript based on the above reasons. As an independent study, our manuscript has more interesting findings than that of Jiang *et al.* (published in *Nature Communications* very recently). Therefore, we believe that our manuscript is deserved to be published in *Nature Communications*.

Comment 2: I believe that the analysis of the ferromagnetism is weak and largely incorrect. The authors find band spin-splitting in the metallic phase, as expected by the Stoner criterion applied to a narrow band related to a 2p shell. This is expected and reasonable. Their analysis of the critical temperature, however, is simply manufactured to get the right answer. Firstly, they map the DFT results on an Ising model. This has no ground, since the system does not possess uniaxial anisotropy (they did not calculate that, and anyway it is expected to be tiny in these organic compounds). The more appropriate mapping would be onto a Heisenberg model, which, of course, includes transverse spin fluctuations. Here the second problem arises. In 2D the Heisenberg model does not have a ferromagnetic phase transition if the coupling is short range, namely $T_C=0$. This is in virtue of the Wagner-Mermin theorem. In a nutshell, the authors map the (probably correct) band structure on the wrong model, just to have the experimental T_C , since a correct mapping would not give any T_C .

Note, in passing, that the T_C for the Ising model does not require any Monte Carlo simulations. It is a known analytical result, $T_C*k_B=2/\ln(1+\sqrt{2})*J$.

Reply 2: We thank the Reviewer for this comment. Based on this comment, we have performed additional DFT calculations and also corrected our model using 2D anisotropic Heisenberg model for T_c calculations [the same method has been successfully applied to other 2D systems, as shown in *Phys. Rev. Lett.* **111**, 106805(2013), *J. Am. Chem. Soc.* **140**, 11519 (2018), *npj Comput. Mater.* **4**, 57 (2018), etc.].

Firstly, we have performed additional DFT calculations with spin-orbit coupling (SOC)

effects to obtain the magnetic anisotropic energy of sp^2 C-COF ($\text{MAE} = E_{\text{out-of-plane}} - E_{\text{in-plane}}$), which is the energy difference between the magnetic states with spins along out-of-plane and in-plane directions. The calculated $\text{MAE} = -0.1$ meV (the reduced MAE coefficient $D = 0.4$ meV), which means that a magnetic easy axis is perpendicular to the COF plane.

Secondly, based on the Reviewer's comment, we have performed the Monte Carlo simulations based on the anisotropic 2D Heisenberg model, which is written as

$$H_M = -\sum_{\langle i,j \rangle} J \vec{m}_i \cdot \vec{m}_j - \sum_i D (m_i^z)^2, \quad (\text{R1})$$

where \vec{m}_i and m_i^z represent the magnetic moment and its z component on the i th site respectively, and $\langle i,j \rangle$ confines the nearest-neighbor (NN) exchange coupling between \vec{m}_i and \vec{m}_j . The J is the exchange integral ($J = -\Delta E / 8m^2$, where $\Delta E = E_{FM} - E_{AFM}$) with the assumption of $|\vec{m}_i| = m$. D is the reduced magnetic anisotropic energy coefficient ($D = -\text{MAE} / m^2$). A 100×100 supercell containing 10000 local magnetic moments are adopted, and each simulation lasts 10^9 loops to relax and 10^9 loops to collect the physical quantities. In each loop, the moment in a random site is rotated to a random direction.

Figure R1 | Calculated average magnetization per site as a function of temperature for monolayer sp^2 C-COF and sp^2 N-COF by Monte Carlo simulations. The transition temperatures (T_c) are marked with dashed vertical lines.

The calculated temperature-dependent average magnetic moment per site by Monte Carlo simulation is shown in Fig. R1. Here, both sp^2C -COF and sp^2N -COF are under a hole doping concentration of $n_h=0.5$ holes/u.c.. It can be seen that the in-plane components of magnetization are zero at arbitrary temperature for both COFs, but their out-of-plane components are nonzero at low temperature. Therefore, although the MAE is small, it is very important to create a nonzero T_c of magnetism. The calculated T_c is 5.3 K (close to 8.1 K of experiments) and 1.7 K for sp^2C -COF and sp^2N -COF, respectively. As explained in the main text, the difference of T_c is due to the different bandwidths of top valence bands in these two COFs.

Finally, we want to emphasize that the calculated MAE values of sp^2C -COF and sp^2N -COF in our study is very similar to that of Jiang *et al. Nat. Commun.* **10**, 2207 (2019). However, the calculated T_c value in our study is smaller than that of Jiang *et al.*, e.g., 5.3 K vs 9.3 K for sp^2C -COF. This is because Jiang *et al.* used the mean-field solution to the Ising model ($J_{ex} \sim k_B T_c$, where $J_{ex}=0.8$ meV and $k_B=8.617343 \times 10^{-5}$ eV/K) to estimate the T_c ($T_c=9.2836$ K) of the sp^2C -COF, which could be incorrect and overestimate the value of T_c . As also indicated by the Reviewer, the Ising model cannot be applied to estimate the T_c of 2D systems.

Based on the Reviewer's comment, we have changed Fig. 4c with Fig. R1. The above discussions can be found in the third paragraph of Page 9 in the revised manuscript. Meanwhile, we have added the computational details in the Method section in the manuscript.

Question 3: It seems to me that there is no relation between the spin-orbit coupling and the ferromagnetism. Also it is not clear whether or not SO is necessary for the half-metallic state. I don't think this is the case.

Reply 3: We thank the Reviewer for this good comment. Indeed, it seems that there is no strong relation between SOC and ferromagnetism, as the FM (half-metallic) state under (heavy) hole doping can be obtained just by the spin-polarized DFT calculations without any SOC effect. However, as discussed in Reply 2, the SOC effects will determine the magnetic anisotropy, which is critical for the realization of ferromagnetism at a finite temperature in 2D systems.

Based on the Reviewer's comment, we have added the above discussions in the third paragraph of Page 9 in the revised manuscript.

Question 4: The authors should make an estimate (by DFT for instance) of the magnitude of the SO coupling. Is this practically relevant?

Reply 4: We thank the Reviewer for this comment. The SOC effects are extremely

small in both COFs. For example, the SOC-induced MAE is about -0.1 meV based on our DFT calculations. Meanwhile, the SOC-induced nontrivial gap is ~ 0.03 meV for the quantum anomalous Hall (QAH) state in sp^2 C-COF under a heavy hole doping concentration of $n_h=3.0$ holes/u.c.. As discussed above, in spite of the weak SOC effects, it is indispensable to achieve the observable ferromagnetism in experiments. However, the topological properties of COFs could be very difficult to be observed due to the tiny nontrivial gap.

Based on the Reviewer's comment, we have deleted the discussions of unobservable topological properties in sp^2 C-COF. In the second paragraph of page 7 in the revised manuscript, we state that "The calculated SOC effects ($\Delta_2 \sim 0.03$ meV) are negligible in these two COFs, indicating that it could be challenging to observe the nontrivial topological properties (Fig. 2) in these two COFs." Meanwhile, we only keep the discussion of topological properties of distorted Lieb lattice models (Figure 2) in the revised manuscript, which may be achieved in other discovered Lieb lattice materials in the future.

Question 5: The hole concentrations investigated appear to me unrealistically large. It is true that ionic liquid gating can achieve such doping level, but this is usually done on insulators presenting a moderately large gap, which is not the case here.

Reply 5: We thank the Reviewer's comment. The recent experiments show that it is possible to dope a large hole concentration into a 2D metallic system using ionic liquid gating, e.g., Fe_3GeTe_2 [Y. Deng *et al.*, *Nature* **563**, 94-99 (2018)].

However, in order to avoid the confusions, we have moved the original Figure 5 and related discussions to the Supplementary Material as Fig. S10. Meanwhile, in the main text we only mention that "Finally, we predict that at an extremely heavy hole doping concentration ($n_h=3.0$ holes/u.c.), the system can reach a half-metallic Dirac semimetal state, as shown in Fig. S10 in the Supplementary Material" in the second paragraph of Page 10 in the revised manuscript.

Figure R2 | Magnetic phase transitions as a function of hole doping concentrations. (a)

DFT-calculated ΔE ($\Delta E = E_{\text{FM}} - E_{\text{AFM}}$) as a function of n_{h} for monolayer $sp^2\text{C-COF}$. **(b)** n_{h} dependent magnetic phase diagram for monolayer $sp^2\text{C-COF}$, calculated by the MC simulations with anisotropic 2D Heisenberg models.

In addition, we have further explored the possible magnetic phase transitions of $sp^2\text{C-COF}$ as a function of n_{h} . As shown in Fig. 3 in the Main Text, the VB1 of $sp^2\text{C-COF}$ is contributed by the MO of corner ligands, which has a small effective inter-site (ligand) hopping $t_1=0.1$ eV. Meanwhile, the on-site (ligand) U is usually in the range of 0.5~1.5 eV in the organic systems, which is significantly larger than t_1 . The calculated ΔE as a function of n_{h} is shown in Fig. R2a. When $n_{\text{h}}<0.3$ holes/u.c. (yellow region), the magnetic moment on each ligand is too small to evoke a preferred magnetic ordering, giving rise to a paramagnetic (PM) ground state. When $0.4<n_{\text{h}}<0.75$ holes/u.c, the FM state becomes to be more favorable ($\Delta E<0$, green region) according to the Stoner's criterion. Interestingly, when ΔE reaches a maximum value of -5.7 meV at $n_{\text{h}}=0.55$ holes/u.c., it becomes to be reduced gradually and finally reaches $\Delta E=0$ at $n_{\text{h}}=0.75$ holes/u.c. When $0.75<n_{\text{h}}<1.0$ holes/u.c., the effect of Fermi surface nesting plays a dominated role in determining its magnetic ground-states, as shown in Fig. S9 in the Supplementary Material, giving rise to a Néel AFM state (cyan region).

Employing the Monte Carlo simulations based on the anisotropic 2D Heisenberg model, we further show n_{h} -dependent magnetic phase diagram of $sp^2\text{C-COF}$ in Fig. R2b. It can be seen that the FM or AFM phase can be survived at finite temperatures under different n_{h} . Interestingly, our calculated magnetic phase diagram of $sp^2\text{C-COF}$ system agrees well with that by J. E. Hirsch based on a similar model system (mean field approximation level calculations) under the situations of $6t_1<U<10t_1$ [*Phys. Rev. B* **31**, 4403 (1985)]. Remarkably, the T_{c} of FM phase can reach a maximum value of ~5.7 K for the $sp^2\text{C-COF}$ at $n_{\text{h}}\sim 0.55$ holes/u.c., consistent with the experimental measurement of $T_{\text{c}}\sim 8$ K for $sp^2\text{C-COF}$.

We want to emphasize that the more accurate phase diagram may need more complex methods beyond the above mean field approximation, e.g., quantum Monte Carlo [*Rev. Mod. Phys.* **63**, 1 (1991); *Phys. Rev. Lett.* **121**, 117202 (2018)] or dynamical mean-field theory [*Sci. Adv.* **1**, e1500568 (2015); *Phys. Rev. Lett.* **87**, 067205 (2001); *Rev. Mod. Phys.* **78**, 865 (2006)], which is out of the scope of the current study.

We have added Fig. R2 as Fig. 5 in the revised manuscript. Meanwhile, the original Fig. 5 is removed to the Supplementary Material as Fig. S10. The above discussion can be found in Pages 9-10 in the revised manuscript.

Question 6: There are a number of statements, which are simply incorrect. For instance: "The half-metallic gap is estimated to be as large as 0.1 eV in $sp^2\text{C-COF}$, which is sufficiently large for high-temperature spintronic applications."

One cannot have room-temperature applications if the T_C is around 10K !!

Reply 6: We thank the Reviewer for this comment. We have deleted the above statement in the revised manuscript. It is also noted that the original Figure 5 and related half-metal discussions (under heavy hole doping concentration of $n_h=3$ holes/u.c.) are moved to the Supplementary Material as Fig. S10.

Additional Note:

I have now taken a look at the rebuttal of the authors to the first round of review. Reviewer 1 is satisfied by the reply, so that only Reviewer 2 remains critical. Here is my view:

Comment 7: I think the reviewer is incorrect concerning the question of the 3-band model against the 6-band one (Question 1). One can always construct an effective model with a limited number of degrees of freedom, namely two 3-band model describing respectively the conduction and valence band (this is what was done by the authors in the reply to the referee's comment). Such approach fails if there is some non-trivial interaction between the conduction and the valence band (e.g. interaction between the two manifold). This does not seem to be the case here (the authors do not treat that case).

Reply 7: We thank the Reviewer for agreeing with our tight-binding calculations. There are no significant interactions between conduction and valence bands, and the three CBs or VBs can be separately fitted by two sets of parameters of tight-binding model.

Comment 8: I believe that the referee is correct to ask Question 2. The reply from the authors seems compelling to me.

Reply 8: We thank the Reviewer agreeing with our reply. As discussed in Reply 4, we have deleted all the discussions on the topological properties in these two real COF materials, as their nontrivial gaps induced by the SOC effects are too weak (~ 0.03 meV). We only keep the discussion of topological properties of distorted Lieb lattice models (Figure 2) in the revised manuscript, which may be achieved in other discovered Lieb lattice materials in the future.

In summary, we wish the Reviewer could re-evaluate our manuscript. As an independent study (confirmed by the editor), we believe that our manuscript has more interesting findings than that of Jiang *et al.* (recently published in *Nature Communications*). Therefore, we believe that our manuscript is deserved to be

published in *Nature Communications*.

Reviewers' Comments:

Reviewer #3:

Remarks to the Author:

This is my second review of the paper "Realization of Lieb lattice in covalent-organic frameworks with tunable topology and ferromagnetism". I do apologise for the long time taken for the reply, mostly due to overcommitment. I believe that the authors did a thorough work of review, which has significantly improved the manuscript.

I accept the authors' argument about the novelty. Although I think that the two papers are still rather similar to each other, I don't think that the authors should be penalised just for having had an unlucky track of reviews. The editor seems to have the same opinion. It is also true that, with the changes, the new manuscript is certainly more complete than the one published already.

Unfortunately, I still have objections on the analysis of the magnetic phase diagram. The new model proposed by the authors is now in line with the physics of their system and with their DFT calculations. I am glad to see that the system has anisotropy and that the new model used to evaluate T_C is the proper Heisenberg model with uniaxial anisotropy.

However:

1) First of all, the authors call their model the "anisotropic Heisenberg model". This is incorrect. An anisotropic Heisenberg model is when the exchange constants are an actual tensor, which is not the case here. Their model should be referred to as the Heisenberg model with uniaxial anisotropy.

2) The Heisenberg model in 2D, with or without anisotropy, does not have a phase transition at finite temperature. This is very well known and it is the main result of the Mermin-Wagner theorem: Phys. Rev. Lett. 17, 1133 (1966). As such, the authors should not see a critical temperature. However, it is frequent in Monte Carlo simulations to observe the signature of phase transitions (as shown by the authors here). These are due to finite size effects, namely a critical temperature is observed when the temperature is low enough that the correlation length reaches the size of the simulated lattice. This does not mean that the system has undergone a phase transition.

One can check this in two ways:

i) by performing simulations for larger lattices, where one will see the "critical temperature" moving to lower temperature, and eventually in the limit of an infinite simulation cell T_C will vanish (though in fairness, one has to go to really large lattice sizes to see the effect).

ii) One may define a '3D' model, with exchanges in, say xy plane as $J_x = J_y = J$, and exchange in z direction as J_z , with J_z varying to $J_z = 0.0, 0.01J, 0.1J, 0.5J, 1.0J$. With these, calculate thermal average of '2D' magnetization (averaged over one xy plane), and '3D' magnetization, which is usual average over the entire lattice. One would see that the onset of ' T_c ' for 2D and 3D magnetization has a nearly constant ratio as long as J_z is not zero, and 2D T_c collapses when $J_z = 0$.

Having said all that, it is also true that small interlayer interactions (e.g. a tiny exchange interaction in the third dimension), are enough make the system into real 3D Heisenberg model resulting in a proper phase transition. When this is the case the ranking of the critical temperatures follow that of the temperatures calculated in absence of such interlayer interaction, but for finite systems. In other words one can use the simulations performed by the authors to rank the magnetic strength of the different systems (for the different range of parameters), but not to relate the calculated T_C directly to experiments.

I believe that before finalising the publication the authors should address this point, since at present the paper contains a mistake. I suggest to include a discussion as follows: an Heisenberg model is used to simulate the various magnetic phases. Due to finite size effect, this model show a phase transition temperature. Taking such temperature as a proxy to a real T_C , which occurs only if the Hamiltonian is another one (namely there is symmetry-breaking in the Heisenberg model) the authors can conclude that the critical temperature C-COF is higher than that of N-COF. In doing that, they have to drop claims of good agreement with experiments.

With this correction I will be happy to accept the manuscript.

Reply to Reviewer #3:

Comments:

This is my second review of the paper "Realization of Lieb lattice in covalent-organic frameworks with tunable topology and ferromagnetism". I do apologise for the long time taken for the reply, mostly due to overcommitment. I believe that the authors did a thorough work of review, which has significantly improved the manuscript.

I accept the authors' argument about the novelty. Although I think that the two papers are still rather similar to each other, I don't think that the authors should be penalised just for having had an unlucky track of reviews. The editor seems to have the same opinion. It is also true that, with the changes, the new manuscript is certainly more complete than the one published already.

Unfortunately, I still have objections on the analysis of the magnetic phase diagram. The new model proposed by the authors is now in line with the physics of their system and with their DFT calculations. I am glad to see that the system has anisotropy and that the new model used to evaluate T_C is the proper Heisenberg model with uniaxial anisotropy.

However:

1) First of all, the authors call their model the "anisotropic Heisenberg model". This is incorrect. An anisotropic Heisenberg model is when the exchange constants are an actual tensor, which is not the case here. Their model should be referred to as the Heisenberg model with uniaxial anisotropy.

2) The Heisenberg model in 2D, with or without anisotropy, does not have a phase transition at finite temperature. This is very well known and it is the main result of the Mermin-Wagner theorem: Phys. Rev. Lett. 17, 1133 (1966). As such, the authors should not see a critical temperature. However, it is frequent in Monte Carlo simulations to observe the signature of phase transitions (as shown by the authors here). These are due to finite size effects, namely a critical temperature is observed when the temperature is low enough that the correlation length reaches the size of the simulated lattice. This does not mean that the system has undergone a phase transition.

One can check this in two ways:

i) by performing simulations for larger lattices, where one will see the "critical temperature" moving to lower temperature, and eventually in the limit of an infinite simulation cell T_C will vanish (though in fairness, one has to go to really large lattice sizes to see the effect).

ii) One may define a '3D' model, with exchanges in, say xy plane as $J_x = J_y = J$, and

exchange in z direction as J_z , with J_z varying to $J_z = 0.0, 0.01J, 0.1J, 0.5J, 1.0J$. With these, calculate thermal average of '2D' magnetization (averaged over one xy plane), and '3D' magnetization, which is usual average over the entire lattice. One would see that the onset of ' T_c ' for 2D and 3D magnetization has a nearly constant ratio as long as J_z is not zero, and 2D T_c collapses when $J_z = 0$.

Having said all that, it is also true that small interlayer interactions (e.g. a tiny exchange interaction in the third dimension), are enough make the system into real 3D Heisenberg model resulting in a proper phase transition. When this is the case the ranking of the critical temperatures follow that of the temperatures calculated in absence of such interlayer interaction, but for finite systems. In other words one can use the simulations performed by the authors to rank the magnetic strength of the different systems (for the different range of parameters), but not to relate the calculated T_C directly to experiments.

I believe that before finalising the publication the authors should address this point, since at present the paper contains a mistake. I suggest to include a discussion as follows: an Heisenberg model is used to simulate the various magnetic phases. Due to finite size effect, this model show a phase transition temperature. Taking such temperature as a proxy to a real T_C , which occurs only if the Hamiltonian is another one (namely there is symmetry-breaking in the Heisenberg model) the authors can conclude that the critical temperature C-COF is higher than that of N-COF. In doing that, they have to drop claims of good agreement with experiments.

With this correction I will be happy to accept the manuscript.

Reply 1: We thank the Reviewer for his/her positive comments and suggestions.

Following the Reviewer's suggestion:

1. We have changed the descriptions of our model used in MC simulations to "*the 2D Heisenberg Model with uniaxial isotropy*" in Pages 9-12 in the revised manuscript.
2. We agree with the Reviewer that the finite-size effect leads to the finite temperature. Therefore, we have added the statement of "*According to the Mermin–Wagner theorem, for a 2D system there is no true long-range order at finite temperature. Hence our discussion should be constrained on finite-size 2D lattices.*" in the first paragraph of on page 9.
3. We have also added an extra paragraph of "*It is emphasized that a Heisenberg model is adopted to simulate the T_c of various magnetic phases. Due to finite size effect, a series of phase transition temperatures can be obtained. Taking such temperature as a proxy to a real T_c , we can conclude that the critical temperature of*

sp²C-COF is higher than that of sp²N-COF.” in the second paragraph of on page 11.

In summary, we have corrected our statements and added some additional discussions in our manuscript based on the Reviewer’s final comments and suggestions.